# Sugar-mediated regulation of a c-di-GMP phosphodiesterase in *Vibrio cholerae*

Kyoo Heo[1], Young-Ha Park[1], Kyung-Ah Lee[2,3], Joonwon Kim[4], Hyeong-In Ham[1], Byung-Gee Kim[4,5], Won-Jae Lee[2,3] & Yeong-Jae Seok [1]*

Biofilm formation protects bacteria from stresses including antibiotics and host immune responses. Carbon sources can modulate biofilm formation and host colonization in *Vibrio cholerae*, but the underlying mechanisms remain unclear. Here, we show that EIIA$^{Glc}$, a component of the phosphoenolpyruvate (PEP):carbohydrate phosphotransferase system (PTS), regulates the intracellular concentration of the cyclic dinucleotide c-di-GMP, and thus biofilm formation. The availability of preferred sugars such as glucose affects EIIA$^{Glc}$ phosphorylation state, which in turn modulates the interaction of EIIA$^{Glc}$ with a c-di-GMP phosphodiesterase (hereafter referred to as PdeS). In a *Drosophila* model of *V. cholerae* infection, sugars in the host diet regulate gut colonization in a manner dependent on the PdeS-EIIA$^{Glc}$ interaction. Our results shed light into the mechanisms by which some nutrients regulate biofilm formation and host colonization.

[1] School of Biological Sciences and Institute of Microbiology, Seoul National University, Seoul 08826, Republic of Korea. [2] School of Biological Sciences and National Creative Research Initiative Center for Hologenomics, Seoul National University, Seoul 08826, Republic of Korea. [3] Institute of Molecular Biology and Genetics, Seoul National University, Seoul 08826, Republic of Korea. [4] School of Chemical and Biological Engineering and Institute of Molecular Biology and Genetics, Seoul National University, Seoul 08826, Republic of Korea. [5] Institute of Engineering Research, Seoul National University, Seoul 08826, Republic of Korea. *email: yjseok@snu.ac.kr

Most bacterial species can form biofilms, which are multicellular communities embedded in a self-produced polymeric matrix. The ability of pathogenic bacteria to form biofilms facilitates their colonization and persistence in the host due to the evasion of the immune response and increased resistance to many antimicrobials[1,2]. In addition, it has been reported that dispersed cells exhibit acute virulence and efficient transmission to other hosts, indicating that an appropriate transition to planktonic cells from these sessile lifestyles is a crucial factor in their pathogenicity[3–6]. Thus, it is conceivable that sophisticated mechanisms should have evolved for pathogenic bacteria to regulate the transition between these two lifestyles in response to various environmental cues[7].

It has been shown that glucose and some other sugars in the environment induce the multilayer biofilm formation and the sugars promoting biofilm formation are substrates of the phosphoenolpyruvate (PEP):carbohydrate phosphotransferase system (PTS) in *V. cholerae*[7]. This multi-component system mediates the transport of various sugars including glucose in *V. cholerae*, and these PTS sugars are concomitantly phosphorylated during transport[8,9]. The PTS consists of two general components, enzyme I (EI) and histidine phosphocarrier protein (HPr) that participate in the transport of most PTS sugars, and several sugar-specific components collectively known as enzyme IIs (EIIs). EI and HPr transfer a phosphoryl group from PEP to EII components, which finally phosphorylate PTS carbohydrates during their translocation across the membrane.

In addition to its role in carbohydrate transport and phosphorylation, the PTS acts as an efficient signal transduction system, which can sense the availability of carbohydrates in the environment and thereby regulates various cellular functions[10]. Phosphorylation of the PTS components usually increases in the absence and decreases in the presence of a PTS carbohydrate such as glucose[11,12]. Depending on their phosphorylation state, the PTS components regulate various sugar-related phenotypes by interacting with their cognate partners[13]. Although the effects of the PTS on biofilm formation have been reported in several studies[9,14,15], the regulation mechanism of biofilm formation by the PTS has not been fully elucidated.

In enteric bacteria such as *Escherichia coli*, EIIA^Glc (encoded by *crr*) is known to play multiple regulatory roles: dephosphorylated EIIA^Glc inhibits non-PTS sugar permeases, such as lactose permease and stimulates the fermentation/respiration switch protein FrsA[13,16], whereas only phosphorylated EIIA^Glc stimulates adenylate cyclase and thus increases the concentration of cAMP[17], which is known to suppress biofilm formation in *V. cholerae*[18–20]. An interesting difference between the *E. coli* and *V. cholerae* PTS lies in the substrate specificity of EIIA^Glc. While EIIA^Glc is specific for glucose in *E. coli*, its *V. cholerae* ortholog is shared among several PTS sugars, such as *N*-acetylglucosamine, trehalose, and sucrose as well as glucose[21]. Therefore, it could be assumed that EIIA^Glc may have distinct regulatory roles in response to several PTS sugars depending on the species. Unlike in *E. coli*, dephosphorylated EIIA^Glc of *Vibrio vulnificus* inhibits flagella assembly and hence motility, allowing it to efficiently consume a preferable sugar in the environment[22,23]. This regulation of the transition from a motile to non-motile lifestyle mediated by EIIA^Glc might also act on biofilm formation processes on the abiotic or biotic surfaces.

The regulatory functions of EIIA^Glc on biofilm formation were suggested in *V. cholerae* in previous studies. EIIA^Glc was shown to interact with MshH, a homolog of *E. coli* CsrD[14], and a following study showed that dephosphorylated EIIA^Glc activates CsrB/C turnover and increases the amount of CsrA[24], which is known to be a negative regulator of biofilm formation in several bacterial species[25,26]. It was also reported that cAMP, the reaction product

of adenylate cyclase which is regulated by EIIA^Glc, and its receptor protein (CRP) directly and indirectly represses the expression of the diguanylate cyclase CdgA, which positively regulates biofilm formation in the *V. cholerae* C1552 strain[20,27]. While the exogenous addition of cAMP represses the biofilm formation also in the *V. cholerae* MO10 strain, EIIA^Glc was shown to activate biofilm formation in the presence of exogenous cAMP in this strain[28]. Collectively, these findings suggest the additional regulatory role of EIIA^Glc on biofilm formation.

Here, we explored the molecular mechanism of how the biofilm formation is affected by carbon sources in *V. cholerae*. We find that EIIA^Glc interacts with and modulates the activity of a c-di-GMP phosphodiesterase (PDE) depending on its phosphorylation state and thereby regulates biofilm formation in the aquatic and host environment. We propose that EIIA^Glc functions as a PTS sugar-responsive regulator of the c-di-GMP-signaling pathway and determines whether to disperse from or stay in the biofilm in response to carbohydrates.

## Results

**EIIA^Glc interacts with an EAL domain-containing protein in *V. cholerae*.** In many bacteria, EIIA^Glc has been implicated in various sugar-dependent regulatory functions[13]. While it was suggested that EIIA^Glc participates in the regulation of biofilm formation in the presence of PTS carbohydrates in *V. cholerae*[9,15], no operative mechanisms were yet offered. To examine the effects of EIIA^Glc on biofilm formation, we measured biofilm production in the presence and absence of glucose. While wild-type *V. cholerae* had a higher level of biofilm formation in the presence of glucose than its absence, which was consistent with previous reports[29,30], a *crr*-deficient mutant exhibited no difference between the two conditions (Fig. 1a). Interestingly, while the *crr* mutant had a similar growth (Supplementary Fig. 1), this mutant exhibited a higher level of biofilm formation compared to the wild-type strain in LB medium (Fig. 1a), which is contrary to a previous study[9]. The exogenous addition of cAMP did not significantly alter the sugar effect, indicating that the regulation of biofilm formation by EIIA^Glc is independent of cAMP. These results led us to search for the regulator of biofilm formation, which transduces the sugar signal by directly interacting with EIIA^Glc in *V. cholerae*.

To find a new interaction partner of EIIA^Glc, we performed ligand fishing experiments using hexahistidine-tagged EIIA^Glc (His-EIIA^Glc) as bait[22]. Total proteins extracted from wild-type *V. cholerae* O1 biovar El Tor N16961 cells grown overnight at 37 °C were mixed with TALON metal-affinity resin in the absence and presence of purified His-EIIA^Glc. His-EIIA^Glc was dephosphorylated by adding glucose or phosphorylated by adding PEP to the mixtures. After several washes, total proteins bound to the resins were eluted with 200 mM imidazole and analyzed by SDS–PAGE and staining with Coomassie brilliant blue R (Fig. 1b). In repeated experiments, we could find three protein bands migrating with apparent molecular masses of approximately 100, 90, and 60 kDa, respectively, that were significantly and reproducibly enriched in the fraction containing both His-EIIA^Glc and glucose (lane 2). Peptide mapping of these proteins following in-gel tryptic digestion revealed that the protein band migrating at ~100 kDa corresponded to VC2072, an ortholog of the insulin-degrading enzyme IDE (VcIDE)[31], and the band at ~60 kDa to VCA1085, an ortholog of the flagella assembly protein FapA (VcFapA)[22]. Since IDE and FapA were already reported to interact with EIIA^Glc in *V. vulnificus*[22,31], the elution of the two proteins only in the fraction containing His-EIIA^Glc indicates the reliability of this ligand fishing method (compare lanes 1 and 2). Interestingly, the band at ~90 kDa was identified as a putative

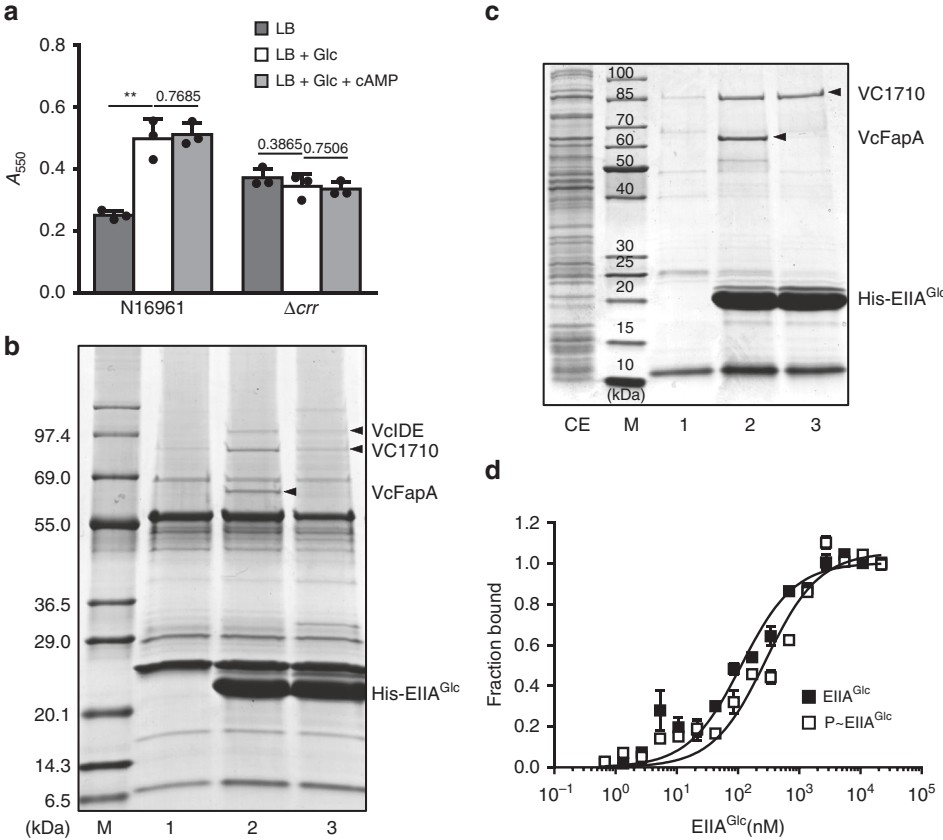

**Fig. 1** Specific interaction of VC1710 with EIIA$^{Glc}$. **a** The biofilm-forming activity of wild-type *V. cholerae* N16961 and an otherwise isogenic Δ*crr* mutant was measured in LB alone or supplemented with glucose or glucose and cAMP, as indicated. Biofilm formation was assessed following the static growth of *V. cholerae* cells for 23 h using a crystal violet staining method[61,62]. The stained biofilm formation was determined at 550 nm. Statistical significance was assessed using Student's *t*-test (*p*-values > 0.05 were presented, \*\**p*-value < 0.01). Shown are the means and SD (*n* = 3, independent measurements). **b** Ligand fishing experiment was carried out to find proteins interacting with His-EIIA$^{Glc}$. Crude extract prepared from *V. cholerae* O1 biovar El Tor N16961 cells was mixed with buffer A (lane 1) or 100 μg of purified His-EIIA$^{Glc}$ (lanes 2 and 3). The extract containing His-EIIA$^{Glc}$ was supplemented with either 2 mM glucose to dephosphorylate EIIA$^{Glc}$ (lane 2) or 2 mM PEP to phosphorylate EIIA$^{Glc}$ (lane 3). Each mixture was subjected to TALON metal affinity chromatography and proteins bound to the column were analyzed as described in the "Methods" section. **c** A mixture of *E. coli* cell lysates expressing recombinant VC1710 and VCA1085 (*V. cholerae* FapA) (lane CE) was mixed with buffer A or His-EIIA$^{Glc}$ and subjected to TALON metal affinity chromatography as in panel **b**. **d** The binding affinities of VC1710 with either dephosphorylated (black square) or phosphorylated EIIA$^{Glc}$ (white square) were measured using NanoTemper Monolith NT.115$^{pico}$. The dissociation constants (*K*$_d$) of VC1710 complexed with dephosphorylated and phosphorylated EIIA$^{Glc}$ (P~EIIA$^{Glc}$) were obtained from three technical replicates. Shown are the means and SD (*n* = 3, independent measurements).

EAL domain-containing c-di-GMP PDE encoded by *vc1710*. The EAL domain is responsible for degrading bis-(3′–5′)-cyclic diguanosine monophosphate (c-di-GMP) to pGpG[32], which is known to repress biofilm formation and induce biofilm dispersion. Thus, we assumed that the interaction of EIIA$^{Glc}$ with this EAL domain protein might be implicated in biofilm formation as a response to the sugar signal.

Interactions of a PTS component with its target proteins are usually dependent on the phosphorylation state of that PTS component. While the specific interaction between IDE and EIIA$^{Glc}$ was reported to be independent of the phosphorylation state of EIIA$^{Glc}$[31], FapA was shown to interact only with dephosphorylated EIIA$^{Glc}$ in *V. vulnificus*[22]. In accordance with these previous studies, VcFapA was not co-eluted with His-EIIA$^{Glc}$ in the fraction incubated with PEP (lane 3 in Fig. 1b), while the VcIDE band was clearly detected in the eluted fraction containing phosphorylated His-EIIA$^{Glc}$. Since VC1710 band was detected in the two fractions incubated with EIIA$^{Glc}$ and either glucose or PEP, we assumed that VC1710 interacts with both dephosphorylated and phosphorylated EIIA$^{Glc}$. To validate the ligand-fishing data, His-EIIA$^{Glc}$, EI, and HPr was mixed with glucose or PEP, and the phosphorylation state of His-EIIA$^{Glc}$ was confirmed by its mobility shift in an SDS–PAGE gel first[33]. Then a mixture of *E. coli* cell extracts expressing recombinant VC1710 and VcFapA (lane CE) was added and subjected to protein affinity pull-down assays to determine their interaction with either form of EIIA$^{Glc}$ (lanes 2 and 3 in Fig. 1c). As expected, while VcFapA interacted only with dephosphorylated EIIA$^{Glc}$, VC1710 appeared to interact with both the dephosphorylated and phosphorylated forms of EIIA$^{Glc}$.

For quantitative analysis of the binding affinity of VC1710 for EIIA$^{Glc}$, the dissociation constants (*K*$_d$) of VC1710 complexed with dephosphorylated and phosphorylated EIIA$^{Glc}$ (P~EIIA$^{Glc}$) were measured by microscale thermophoresis (MST) experiments. VC1710 showed a slightly higher affinity toward the dephosphorylated form of EIIA$^{Glc}$ (*K*$_d$ = 114.1 ± 13.7 nM) than toward phosphorylated EIIA$^{Glc}$ (286.7 ± 39.5 nM) (Fig. 1d). This tight interaction of VC1710 with EIIA$^{Glc}$ led us to assume that the physiological form of VC1710 exists as a complex with EIIA$^{Glc}$ and the activity of VC1710 can be influenced by the phosphorylation state of EIIA$^{Glc}$, as exemplified by the interaction of adenylate cyclase with EIIA$^{Glc}$ in *E. coli*[17].

**EIIA$^{Glc}$ regulates c-di-GMP PDE activity of VC1710.** An analysis of the primary structure of VC1710 indicated that this protein consists of a PAS-sensing domain in the N-terminus and an EAL domain predicted to encode a c-di-GMP PDE in the C-terminus[34]. It has also been previously reported that VC1710 binds c-di-GMP[35], suggesting its function in c-di-GMP metabolism. To determine whether VC1710 possesses PDE activity, we purified VC1710 and performed an in vitro PDE activity assay by reverse-phase HPLC. The concentration of remaining c-di-GMP decreased with a concomitant increase in the product pGpG after incubation with VC1710, and we could therefore confirm that VC1710 exhibited the predicted c-di-GMP hydrolytic activity (Fig. 2a, b). Then, to investigate if the PDE activity of VC1710 was affected by its interaction with EIIA$^{Glc}$, dephosphorylated or phosphorylated EIIA$^{Glc}$ was added to the reaction mixture. The phosphorylation states of EIIA$^{Glc}$ could be successfully modulated by incubation with EI, HPr, and PEP, as shown in the SDS–PAGE analysis (Fig. 2c). When phosphorylated EIIA$^{Glc}$ was mixed with VC1710, the PDE activity was higher than that in the absence of EIIA$^{Glc}$. However, no c-di-GMP was digested in the reaction incubated with dephosphorylated EIIA$^{Glc}$, suggesting that dephosphorylated EIIA$^{Glc}$ inactivates the PDE activity of VC1710. To elucidate the kinetic properties of the VC1710 protein, we determined $K_m$ and $V_{max}$ values based on three independent activity assays. When c-di-GMP was reacted with VC1710 alone, the $K_m$ value was ~4.58 µM. However, when phosphorylated EIIA$^{Glc}$ was added to the VC1710 reaction mixture, the $K_m$ value for c-di-GMP decreased to 1.30 µM (Supplementary Fig. 2). Given that the $K_d$ values for c-di-GMP-binding proteins ranged up to several µM and the $K_m$ values of PDEs ranged from 60 nM to several µM[36], the $K_m$ values of VC1710 for c-di-GMP appear to be biochemically and physiologically relevant to the control of the c-di-GMP level and thereby related phenotypes in vivo. To exclude the possibility of carrier protein effect, we performed the same experiment without EIIA$^{Glc}$ or in the presence of the same amount of BSA, and no measurable effect was detected (Supplementary Fig. 3). The analytical western blot data (Supplementary Fig. 4) shows that the number of the EIIA$^{Glc}$ protein in a *V. cholerae* cell is more than 100 times higher than that of the VC1710 protein. We, therefore, conclude that VC1710 is always present in a complex form with EIIA$^{Glc}$ in the cell and that the activity of VC1710 is entirely dependent on the phosphorylation state of EIIA$^{Glc}$. As the phosphorylation state of EIIA$^{Glc}$ was altered in response to the type of sugars[11], we named VC1710 as PdeS (for Sugar-responsive PDE).

From the MST assay, we could measure the dissociation constant of the PdeS/c-di-GMP complex to be ~8.7 µM (Fig. 2d), which is comparable with the $K_m$ value of PdeS toward c-di-GMP. Interestingly, however, we could not detect any interaction of c-di-

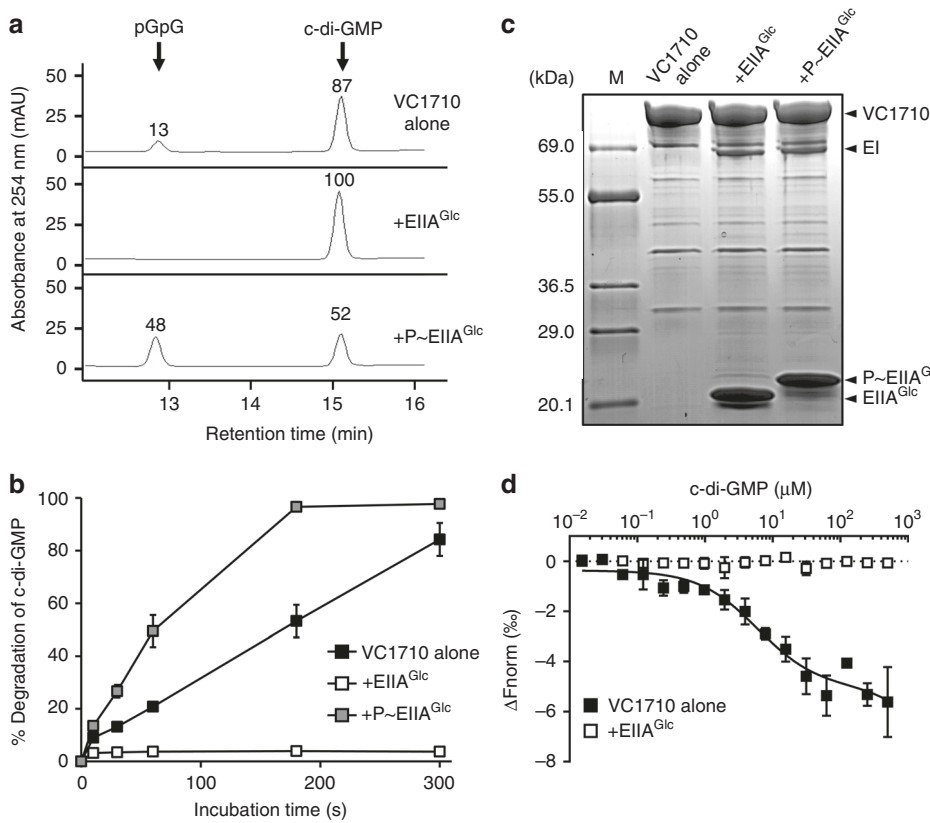

**Fig. 2** EIIA$^{Glc}$ modulates the PDE activity of VC1710. **a** The c-di-GMP phosphodiesterase (PDE) activity of VC1710 (4.1 µM) was assayed in a reaction mixture containing 20 µM of c-di-GMP and 0.4 µM of EI and HPr in the absence or presence of purified EIIA$^{Glc}$ (17.1 µM). EIIA$^{Glc}$ in the reaction mixture was phosphorylated by adding 2 mM PEP. The reaction mixtures were applied to a Supercosil LC-18-T HPLC column, and the remaining c-di-GMP and produced pGpG were quantified by measuring absorbance at 254 nm. **b** The c-di-GMP hydrolysis activity of VC1710 was assessed in the absence (black square) and presence of EIIA$^{Glc}$ (white and gray squares) as in panel **a** and c-di-GMP and pGpG were analyzed at 0, 10, 30, 60, 180, and 300 s. **c** The phosphorylation state of EIIA$^{Glc}$ in panels **a** and **b** was confirmed by SDS–PAGE and staining with Coomassie brilliant blue R. **d** The binding affinities of PdeS with c-di-GMP in the absence (black square) or presence (white square) of dephosphorylated EIIA$^{Glc}$ were measured using NanoTemper Monolith NT.115$^{pico}$. An excess amount (5 µM) of EIIA$^{Glc}$ was added to the reaction mixture to ensure all the VC1710 existed as a complex with EIIA$^{Glc}$. Shown are the means and SD ($n = 3$, independent measurements).

GMP with PdeS in the presence of dephosphorylated EIIA$^{Glc}$, indicating that PdeS becomes inaccessible to c-di-GMP when it forms a complex with the dephosphorylated form of EIIA$^{Glc}$ (Fig. 2d). Therefore, it could be assumed that the tight binding of dephosphorylated EIIA$^{Glc}$ inactivates the PDE activity of PdeS by blocking the accessibility of c-di-GMP to the active site.

**EIIA$^{Glc}$ regulates the activity of PdeS in vivo.** To test whether PdeS can also digest c-di-GMP in the cell, we compared intracellular concentrations of c-di-GMP between the wild-type and a *pdeS*-deficient mutant strain (Δ*pdeS*) of *V. cholerae* N16961 (Fig. 3a). Because EIIA$^{Glc}$ in cells growing in LB medium is mostly in its phosphorylated form (Fig. 4a), which stimulates the PDE activity of PdeS, the cells grown in the buffered LB medium were used to extract c-di-GMP. Measurement using LC–MS/MS revealed that Δ*pdeS* cells had an ~2.5-fold higher concentration of c-di-GMP than wild-type cells as expected. In addition, the Δ*pdeS* strain had a higher expression level of the *Vibrio* polysaccharide synthesis (*vps*) operon and thus formed significantly more biofilm than the wild-type strain, which is consistent with previous reports showing that c-di-GMP induces biofilm formation[37,38] (Fig. 3a, b; Supplementary Fig. 5a). We found that an alanine substitution mutant in the EAL domain of PdeS (PdeS(E450A)) resulted in the complete abolishment of the PDE activity (Supplementary Fig. 3). While the Δ*pdeS* strain carrying an expression vector for wild-type PdeS exhibited similar levels of biofilm formation and *vps* expression with those of the wild-type strain, the mutant strain expressing PdeS(E450A) did not complement these phenotypic changes (Supplementary Figs. 5b and 6). Thus, we concluded that PdeS regulates biofilm formation through its PDE activity.

Then, to confirm whether the regulation of the PdeS activity by EIIA$^{Glc}$ also operates in *V. cholerae* cells, we constructed a dephosphomimetic mutant (H91A) of *crr* encoding EIIA$^{Glc}$ on the chromosome and performed biofilm formation assays (Supplementary Fig. 7a). While we observed that the dephosphomimetic *crr* mutant showed increased biofilm formation compared to the wild-type strain, this stimulatory effect was not observed in a *pdeS* deletion mutant. EIIA$^{Glc}$ is shared for several membrane-bound enzyme IIBCs including those specific for glucose, *N*-acetylglucosamine, sucrose, and trehalose in *V. cholerae*[9] and phosphorylatable EIIA$^{Glc}$ is indispensable for a variety of physiological processes including the regulation of global transcription factors such as CRP and Mlc. Therefore, we assume that the phenotype of this chromosomal *crr* mutant might be due to indirect pleiotropic effects of the mutation. For this reason, we tried to see the effect of EIIA$^{Glc}$ dephosphorylation on biofilm formation by increasing the level of the dephosphomimetic mutant of EIIA$^{Glc}$, while minimizing the perturbation of the overall PTS activity. Therefore, we compared biofilm formation and the intracellular level of c-di-GMP of the wild-type strain carrying an expression vector for wild-type EIIA$^{Glc}$ with the wild-type strain carrying an expression vector for EIIA$^{Glc}$(H91A), and observed increased biofilm formation and the c-di-GMP level in the latter strain (Fig. 3c; Supplementary Fig. 7), which is in accordance with the result obtained by the chromosomal mutation of the *crr* gene. Together our data show that dephosphorylated EIIA$^{Glc}$ inactivates the PDE activity of PdeS and thereby increases the c-di-GMP level in vivo. Thus, it could be assumed that EIIA$^{Glc}$ can modulate biofilm formation by controlling the c-di-GMP level through direct interaction with PdeS.

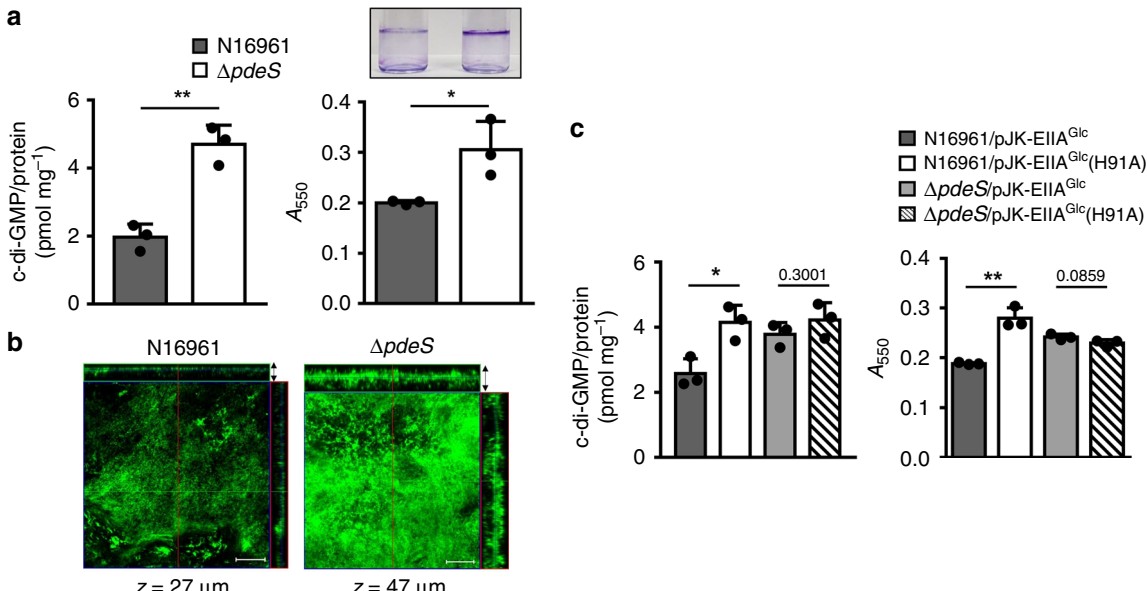

**Fig. 3** PdeS regulates the c-di-GMP level and biofilm formation. **a** The intracellular c-di-GMP concentration and the biofilm-forming activity of wild-type N16961 and a Δ*pdeS* strain were measured in the buffered LB medium. The concentration of c-di-GMP extracted from the *V. cholerae* cells was determined using LC–MS/MS, and normalized with total protein contents. Biofilm formation was assessed following static growth of *V. cholerae* cells for 23 h using a crystal violet staining method[61,65]. The biofilm formation was determined at 550 nm. **b** Biofilm formation was visualized using confocal laser scanning microscopy. Scale bar: 50 μm. **c** The effect of EIIA$^{Glc}$ on the c-di-GMP hydrolysis activity of PdeS was assessed in vivo. The intracellular c-di-GMP concentration and the biofilm-forming activity of wild-type and Δ*pdeS* *V. cholerae* strains harboring a pBAD-based expression vector for either wild-type EIIA$^{Glc}$ or dephosphomimetic mutant EIIA$^{Glc}$(H91A) were determined in the presence of 0.1% arabinose as described in panel **a**. Statistical significance was assessed using Student's *t*-test (*p*-values > 0.05 were presented, **p*-value < 0.05, ***p*-value < 0.01). Shown are the means and SD (*n* = 3, independent measurements).

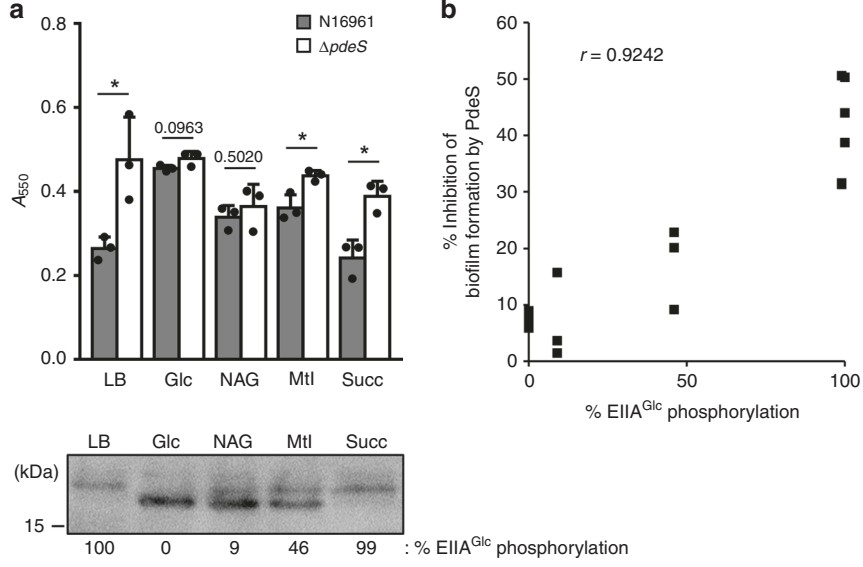

**Fig. 4** PdeS regulates biofilm formation depending on carbon sources. **a** The level of biofilm formation was compared between wild-type N16961 and a $\Delta pdeS$ strain in the presence of various carbon sources in buffered LB medium as in Fig. 3. At the same time, the phosphorylation state of EIIA[Glc] was determined by western blot using anti-EIIA[Glc] mouse serum after SDS–PAGE. Statistical significance was assessed using Student's $t$-test (*$p$-value < 0.05, $p$-values > 0.05 were presented). Glc, glucose; NAG, N-acetylglucosamine; Mtl, mannitol; Succ, Succinate. **b** The inhibitory effect of PdeS on the biofilm formation was plotted as a function of EIIA[Glc] phosphorylation. Based on the data in panel **a**, the degree of biofilm formation of wild-type cells was divided by that of the $\Delta pdeS$ mutant cells. The percentage of biofilm formation inhibition by PdeS was then calculated by subtracting the resultant value from 1 and multiplying by 100, then plotted as a function of the percentage of phosphorylated EIIA[Glc] over total EIIA[Glc]. The correlation between the sugar-mediated phosphorylation state of EIIA[Glc] and biofilm formation was assessed using Pearson's correlation coefficient ($r = 0.9242$, $p < 0.005$). Shown are the means and SD ($n = 3$, independent measurements).

**EIIA[Glc] regulates biofilm formation depending on carbon sources**. PTS components including EIIA[Glc] can have a different phosphorylation state depending on the availability of PTS substrates. As the PDE activity of PdeS and thus the intracellular concentration of c-di-GMP are modulated depending on the phosphorylation state of EIIA[Glc], we examined whether biofilm formation can be influenced by carbohydrates encountered by *V. cholerae* in their environmental niches. Thus, we performed a biofilm formation assay in the presence of various carbon sources in LB medium. To exclude the effect of the pH decrease due to fermentation of sugar, we buffered LB medium with 40 mM potassium phosphate, pH 7.0. At the same time, we determined the in vivo phosphorylation state of EIIA[Glc] in each medium (Fig. 4a). Consistent with previously reported results in *E. coli* and *V. vulnificus*[11,22], EIIA[Glc] was mostly dephosphorylated in glucose-containing medium, while mostly phosphorylated in LB medium without additional carbon source. We could also find a significant positive correlation between the degree of EIIA[Glc] phosphorylation and the inhibitory effect of PdeS on biofilm formation (Fig. 4b). These data imply that the modulation of biofilm formation by PdeS is dependent on the phosphorylation state of EIIA[Glc], which is determined by the environmental carbon source.

**EIIA[Glc] regulates biofilm formation and host gut colonization**. *V. cholerae* is associated with various hosts, including arthropods, insect eggs, and unicellular eukaryotes as well as humans[39]. Recently, the interaction between *V. cholerae* and the fruit fly *Drosophila melanogaster* has been intensively studied, since this arthropod model acts as a disease reservoir in nature[40] and is also simple but has similar physiological features and anatomical structures with mammalian infection models[41]. It has been reported that *Vibrio* exopolysaccharide (VPS)-dependent biofilm formation is indispensable for attachment and colonization in the

*Drosophila* intestine in a quorum sensing-defective *V. cholerae* strain[42,43]. Therefore, we assumed that VPS might also play a critical role in the intestinal colonization in the N16961 strain which carries a natural frame-shift mutation in the *hapR* gene encoding the quorum-sensing master regulator HapR. As the expression of the *vps* operon was controlled by c-di-GMP, which is the substrate of PdeS, we assumed that PdeS might also play a role in the regulation of intestinal colonization. Since mannitol transport by *V. cholerae* requires mannitol-specific EII but not EIIA[Glc] in spite of its structural similarity with glucose, the phosphorylation state of EIIA[Glc] is different in the presence of the two PTS sugars (Fig. 4a). Therefore, we chose glucose and mannitol as representative sugars leading to dephosphorylation and ~50% phosphorylation of EIIA[Glc], respectively, to evaluate the EIIA[Glc] phosphorylation state-dependent regulation of the colonization efficiency of *V. cholerae*. To investigate whether a sugar-dependent change in EIIA[Glc] phosphorylation influences bacterial colonization in the intestine through the modulation of PdeS activity, flies were fed 5% sugar (glucose or mannitol) solution containing ~$10^6$ cells $\mu l^{-1}$ of the *V. cholerae* N16961 strain for 24 h, and subsequently fed the same sugar solution without bacteria for 9 h. These flies were then surface-sterilized, homogenized in 1 ml of PBS buffer and spread on an agar plate for the quantification of the colonization of *V. cholerae* in vivo by measuring colony-forming units (CFUs). While the CFU of the $\Delta pdeS$ strain was not affected by the sugar source, the CFU of the wild-type strain was significantly lower in flies fed mannitol compared to flies fed glucose (Fig. 5a). In flies fed mannitol, the wild-type strain gave significantly lower CFU levels compared to the $\Delta pdeS$ strain. Therefore, we could assume that, in the presence of mannitol, phosphorylated EIIA[Glc] stimulates the PdeS activity to decrease the c-di-GMP level and thereby biofilm formation.

To determine whether the effect of PdeS on intestinal colonization is mediated by the regulation of biofilm formation,

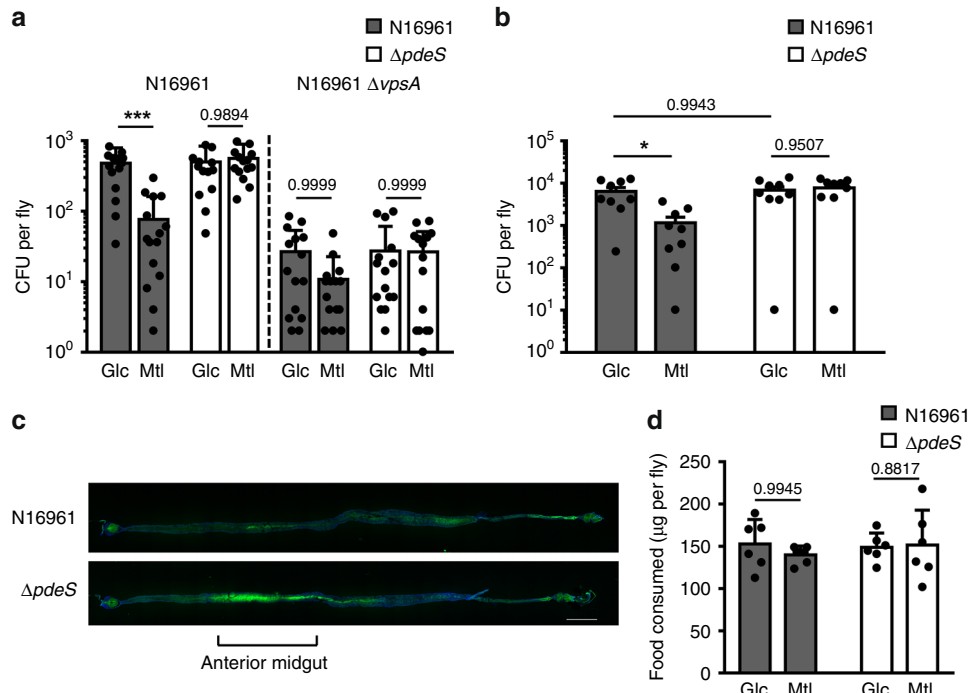

**Fig. 5** PdeS regulates intestinal colonization depending on the host diet. **a** Flies were orally administered a 5% sugar solution containing the indicated *V. cholerae* strains for 24 h, and fed a sterile sugar solution for another 9 h. The homogenate of the surface-sterilized fly was spread on a selective medium for *V. cholerae* and colony-forming units (CFU) were measured. **b** Intestinal colonization was assessed for a shorter infection period. Flies were orally administered a 5% sugar solution containing the indicated *V. cholerae* strains for 2 h, and fed a sterile sugar solution for another 2 h. **c** The *V. cholerae* colonization on the alimentary canal was visualized using confocal laser scanning microscopy. Flies were exposed to *V. cholerae* strains harboring a GFP expression vector in the presence of 5% mannitol for 2 h and, after incubation in a sterile medium for another 2 h, the whole intestines were dissected and visualized. Shown are the representative images from three independent experiments on different days. Scale bar: 500 μm. **d** Food consumption was measured by dye uptake using FD&C Blue #1 dye during bacterial infection[64]. Shown are the means and SD ($n = 6$, independent measurements). Data was analyzed using a two-way ANOVA followed by Tukey's post hoc test (*$p$-value < 0.05, ***$p$-value < 0.005, $p$-values > 0.05 were presented).

we repeated the colonization experiment with Δ*vpsA* and Δ*vpsA pdeS* mutants (Fig. 5a). As reported previously[42,43], mutation of *vpsA* resulted in a significantly reduced (~5%) intestinal colonization in flies compared to the wild-type strain. Interestingly, the inhibitory effect of PdeS on intestinal colonization was not seen in this *vpsA*-mutant strain. Thus, it could be assumed that VPS-dependent biofilm formation is important for the regulation of intestinal colonization by PdeS.

Then, to assess whether PdeS affects the earlier stage of *V. cholerae* infection of the *Drosophila* intestine, we infected flies orally with 5% sugar solution containing ~$10^8$ cells μl$^{-1}$ of wild-type *V. cholerae* N16961 or its *pdeS* mutant for 2 h. After feeding flies with bacteria-free 5% sugar solution for another 2 h and surface sterilization, we quantified the intestinal colonization of *V. cholerae* by measuring CFUs (Fig. 5b). While the CFU of the Δ*pdeS* strain showed no difference depending on the sugar source, the CFU of the wild-type strain was significantly lower in flies fed mannitol compared to flies fed glucose. Therefore, our data indicates that PdeS may regulate the initial phase of intestinal colonization of *V. cholerae*.

To determine which compartment(s) *V. cholerae* colonizes and to examine the physiological relevance of the host diet-dependent colonization, flies were exposed to the *V. cholerae* N16961 strain carrying a GFP expression vector in the presence of 5% mannitol, and their colonization on the whole alimentary canal was visualized (Fig. 5c). Although the amounts of ingested bacteria (as estimated by the ingested amounts of bacteria-containing sugar solutions) were similar between wild-type and Δ*pdeS* strains (Fig. 5d), more intensive colonization was detected in flies infected with the Δ*pdeS* strain than in flies infected with the

wild-type strain, especially in the anterior midgut, which is the main site of digestion and nutrient absorption[44] in our infection condition (Fig. 5c). This result is consistent with the CFU data (Fig. 5b), implying that PdeS modulates the colonization efficiency of *V. cholerae* depending on the host diet, which is consistent with its sugar-dependent regulation of c-di-GMP and biofilm formation.

## Discussion

The pathogen *Vibrio cholerae*, like other bacteria, has various niches including natural seawater, copepod, chironomid egg masses, and the human host[39]. The ability to sense and respond to each environment is the crucial factor for adaptation and propagation of bacteria. In many bacteria, the PTS is one of the sensory systems that regulate multiple metabolic pathways in response to the availability of carbon or nitrogen nutrients in diverse environments[45]. Previous studies suggested that the PTS in *V. cholerae* also has regulatory functions in biological processes, such as biofilm formation[14,28], colonization in the mammal host[9], chitin utilization, and natural competence[46]. Because most of these regulations are mediated by protein–protein interactions[13], the characterization of the interaction network of the PTS is vital for understanding the orchestrated cellular responses to environmental changes. In this study, we found a new interaction partner of EIIA$^{Glc}$, the c-di-GMP PDE PdeS which hydrolyzes the ubiquitous bacterial second messenger c-di-GMP. As the c-di-GMP-signaling pathway is involved in many behaviors, such as biofilm formation, motility, and toxin production, this interaction is expected to be an important key to reveal many nutrient-related phenotypes.

Many studies suggested that c-di-GMP induces the transition from motile to sessile mode by binding to and regulating various receptors involved in biofilm formation[27,47]. Biofilm matrix serves as a barrier against various environmental stresses, which enables bacteria to prosper as long as sufficient nutrients are available. However, once nutrients become scarce, bacterial cells detach and disperse from this community by reducing the production of this polymeric matrix. Several studies have reported that nutrient deprivation decreases biofilm formation and induces biofilm dispersion by decreasing intracellular c-di-GMP[48–50]. The interaction between EIIA$^{Glc}$ and PdeS is expected to be one of these regulatory mechanisms.

It is notable that modulation of c-di-GMP hydrolytic activity of PdeS by EIIA$^{Glc}$ is comparable to the regulation of cAMP synthetic activity of adenylate cyclase[17]. In several bacteria, only the phosphorylated form of EIIA$^{Glc}$ is known to stimulate cAMP synthesis[13,17,22]. Similarly, our data in this study show that dephosphorylated EIIA$^{Glc}$ strongly inhibits c-di-GMP hydrolysis, whereas its phosphorylated form stimulates c-di-GMP hydrolysis in V. cholerae. The copy number of the EIIA$^{Glc}$ protein was reported to be 20,000–30,000, while that of adenylate cyclase to be <10 in E. coli[51]. According to our analytical western blot results, the copy number of EIIA$^{Glc}$ in a V. cholerae cell was estimated to be similar to that in E. coli and that of the PdeS protein to be 29–153 (Supplementary Fig. 4). Because the number of EIIA$^{Glc}$ is significantly higher than that of PdeS and adenylate cyclase, EIIA$^{Glc}$ appears to simultaneously control the amount of c-di-GMP and cAMP, yet in the opposite direction, in V. cholerae. As the cAMP-CRP complex has been reported to inhibit biofilm formation by differentially regulating the expression of several c-di-GMP-metabolizing enzymes[18,20], it was expected that the cAMP pathway would mitigate the regulatory effect of PdeS on biofilm formation. However, the expression level of PdeS was not changed by the sugar type (Supplementary Fig. 8). Moreover, the PdeS-mediated regulation of biofilm formation was quantitatively correlated to the phosphorylation state of EIIA$^{Glc}$ (Fig. 4). Therefore, we concluded that although the intracellular level of cAMP and c-di-GMP was controlled at the same time, the sugar-dependent c-di-GMP-regulatory pathway is hardly affected by the cAMP-signaling pathway in V. cholerae N16961. In addition, it could be assumed that the indirect interplay between the cAMP and c-di-GMP-signaling pathways might play only a minor role, if any, in the sugar-dependent regulation of biofilm formation in this strain. In previous studies, different effects of cAMP–CRP on biofilm formation have been reported among various V. cholerae strains. While biofilm formation was negatively regulated by the cAMP–CRP complex in the C1552 and C6728 strains[20,52], the supplementation of the growth medium with various concentrations of cAMP had no effect on the total growth and biofilm accumulation by a crr mutant of the MO10 strain[28]. Herein, we report that the sugar-dependent regulation of biofilm formation is not affected by cAMP–CRP in the N16961 strain. This may simply be the result of strain differences, as V. cholerae is known to undergo genetic drift in laboratory culture[53].

During the host infection, the intracellular concentration of c-di-GMP in V. cholerae is changed along its infection stage[54]. Also, the c-di-GMP pool fluctuates in response to various external signals, choosing the fittest infection strategy depending on the environmental condition[55]. Thus, the sophisticated regulation of c-di-GMP contents appears to be a prerequisite for a successful propagation and transmission to new hosts throughout the infection cycle. In this respect, our findings provide a new insight into how pathogenic bacteria cope with fluctuating nutritional conditions including those encountered during passage through the intestinal tract of the host.

## Methods

**Bacterial strains, plasmids, and culture conditions**. The bacterial strains and plasmids used in this study are listed in Supplementary Table 1. Construction of the deletion strains was performed as described previously[56]. V. cholerae strains were cultured in Luria-Bertani medium.

**Purification of overexpressed proteins**. While EI, HPr, and EIIA$^{Glc}$ were expressed in a ptsHIcrr-deleted E. coli ER2566 strain, other proteins were expressed in wild-type ER2566 by adding 1 mM IPTG. His-tagged proteins were purified using TALON metal-affinity resin (Takara Bio.) according to the manufacturer's instructions. After His-tagged proteins were eluted with 200 mM imidazole, the fractions containing His-tagged proteins were pooled and concentrated using Amicon Ultracel-3K centrifugal filters (Merck Millipore). To increase the purity of proteins and remove imidazole, the concentrated pool was chromatographed on a Hiload 16/60 Superdex 200 pg column (GE Healthcare) equilibrated with buffer A (25 mM HEPES–NaOH (pH 7.6), containing 100 mM NaCl, 10 mM β-mercaptoethanol, and 10% glycerol).

**Ligand fishing using metal-affinity beads**. Ligand-fishing experiments were performed as described previously with minor modifications to find a new interaction partner of EIIA$^{Glc}$[23,57]. V. cholerae O1 biovar El Tor N16961 cells grown at 37 °C overnight at LB (200 ml) were harvested and resuspended in buffer A. Cells were then disrupted by three passages through a French pressure cell at 8000 psi. After centrifugation at 10,000×g for 20 min at 4 °C, the supernatant was mixed with 100 μg of His-EIIA$^{Glc}$ or buffer A as control in the presence of TALON metal-affinity resin in a 15-ml tube, then incubated at 4 °C for 30 min. His-EIIA$^{Glc}$ was dephosphorylated by adding glucose or phosphorylated by adding PEP to the mixtures. After brief washes with buffer A containing 10 mM imidazole, the bound proteins were eluted with buffer A containing 200 mM imidazole and analyzed by SDS–PAGE using a 4–20% gradient Tris–glycine gel (KOMA biotech) and staining with Coomassie brilliant blue R. Protein bands specifically bound to the His-tagged bait protein were excised from the gel, and in-gel digestion and peptide mapping of the tryptic digests were performed using MALDI-TOF MS[57,58].

**c-di-GMP PDE activity assay using HPLC**. c-di-GMP PDE activity was determined by measuring the remaining c-di-GMP and produced pGpG after the reaction[59]. The reaction contained 20 mM Tris–HCl (pH 8.0), 50 mM NaCl, 5 mM MgCl$_2$, 0.5 mM EDTA, and 20 μM c-di-GMP in addition to PdeS in a total volume of 40 μl. To test the effect of the phosphorylation state of EIIA$^{Glc}$ on PDE activity, EIIA$^{Glc}$ and 0.4 μM of EI and HPr were added in the presence or absence of PEP. The phosphorylation states of EIIA$^{Glc}$ were confirmed by SDS–PAGE and staining with Coomassie brilliant blue R[12]. The enzymatic reaction was started by the addition of c-di-GMP and allowed to proceed at 37 °C. Aliquots of each reaction were taken at appropriate times and reactions were stopped by adding 10 mM CaCl$_2$. These mixtures were boiled for 5 min and centrifuged. Then 20 μl of each supernatant was injected into a Supelcosil LC-18-T column (Sigma-Aldrich) using an Agilent HP1200 HPLC system (Agilent Technology). The separations of pGpG and c-di-GMP were performed by gradient elution at a flow rate of 1.0 ml min$^{-1}$ and chromatograms were recorded at 254 nm. The gradient program was: 0–5 min, isocratic elution with 100 mM potassium phosphate, pH 6.0 (A); 5–13 min, linear gradient to 70% A and 30% methanol (B); 13–16 min, isocratic at B; 16–18 min, linear gradient to A; 18–20 min, isocratic at A.

**Determination of the intracellular concentration of c-di-GMP**. V. cholerae was cultivated in LB medium (100 ml) to OD$_{600}$ ~ 1.0, and centrifuged at 4000×g for 10 min. The cell pellet was resuspended in 500 μl extraction buffer (40% methanol, 40% acetonitrile, and 0.1 N formic acid) and incubated on ice for 15 min. Each sample was subjected to three cycles of freezing/thawing using liquid nitrogen and heat block adjusted to 90 °C. After an additional 15-min incubation on ice, the sample was centrifuged at 16,100×g for 10 min, and 400 μl of the supernatant was dried under vacuum. The lyophilized nucleotides were resuspended in 80 μl of distilled water.

Quantification of the c-di-GMP was carried out using HPLC coupled with triple quadrupole mass spectrometry. The temperatures of the column oven and autosampler were maintained at 25 and 20 °C, respectively. Ten microliters of the resuspended nucleotides were injected to a Hypersil GOLD HILIC column (2.1 × 100 mm, particle size 1.9 μM, pore size 175 Å, Thermo Scientific) using Accela 1250 UPLC$^{TM}$ system (Thermo Fisher Scientific, USA). Separation of c-di-GMP was performed using the following gradient program: solvent A, 20 mM ammonium acetate adjusted to pH 8.0 with ammonium hydroxide; solvent B, acetonitrile (ACN); flow rate 250 μl min$^{-1}$; gradient condition, 0–2 min (15% B), 2–29 min (15–98% B), 29–33 min (98% B), 33–35 min (98–15% B), 35–45 min (15% B). Xanthosine 3′,5′-cyclic monophosphate (cXMP; BioLog) was used as an internal standard. The retention times for c-di-GMP and cXMP were 21.5 and 16.8 min, respectively. The analyte detection was performed using TSQ Quantum Access Max (Thermo Scientific). Tune parameters for TSQ were as follows: capillary temperature, 300 °C; vaporization temperature, 250 °C; sheath gas pressure, 30 psi; aux gas pressure, 10 psi; positive polarity spray voltage, 4.0 kV. The samples were monitored with SRM scan mode. The SRM settings for c-di-GMP and cXMP were

optimized and determined as follows: c-di-GMP: $[M + H]^+$ $m/z$ $691 \rightarrow 152$; collision energy, 38 eV; cXMP: $[M + H]^+$ $m/z$ $347 \rightarrow 153$, collision energy, 18 eV. The measured intracellular concentration of c-di-GMP was normalized with the protein level.

**MST analysis.** The binding affinities of PdeS with EIIA$^{Glc}$ and c-di-GMP were measured using a NanoTemper Monolith NT.115$^{pico}$ instrument[60]. Purified GST-PdeS was labeled with NT-647 using a Monolith protein-labeling kit and used at a concentration of 4.875 nM. Each unlabeled EIIA$^{Glc}$ and c-di-GMP was titrated in 1:1 serial dilutions in MST-binding buffer (25 mM HEPES–NaOH (pH 7.6), 100 mM NaCl, 5 mM β-mercaptoethanol, 0.5 mg ml$^{-1}$ BSA, 0.05 (v/v) % Tween 20), with the highest concentration of EIIA$^{Glc}$ at 21.9 μM and c-di-GMP at 500 μM. To prevent degradation of c-di-GMP by PdeS during the assay, their interaction was examined in the buffer without divalent cations which are indispensable for the PDE activity of PdeS. The measurements were performed at 15% LED power and 30% MST power at 22 °C.

**Measurement of biofilm formation.** Overnight-grown cells were inoculated in LB medium buffered with 40 mM potassium phosphate (pH 7.0) in the absence or presence of sugar and incubated under static conditions in a borosilicate tube at 37 °C for 23 h. After planktonic cells were washed away with PBS, the remaining biofilm-associated cells were stained with 0.1% crystal violet (CV) for 20 min. After rinses with PBS three times, the CV-stained biofilm was solubilized with 95% ethanol and measured at 550 nm[61,62]. Mature biofilm was also visualized as described previously[63] using confocal laser scanning microscopy (LSM700, Zeiss). *V. cholerae* strains constitutively expressing green fluorescent protein (GFP) were used for biofilm imaging.

**Determination of the phosphorylation state of EIIA$^{Glc}$.** The phosphorylation state of EIIA$^{Glc}$ was determined as described previously[12,33] with some modifications. A 0.2-ml aliquot of cell culture was quenched at OD$_{600}$ ~ 0.5 by adding 20 μl of 5 M NaOH followed by vortexing for 15 s, and then 150 μl of 3 M sodium acetate (pH 5.3) and 0.9 ml of ethanol were sequentially added. After incubated at −80 °C for 30 min, each sample was centrifuged at 10,000×g at 4 °C for 30 min, and the pellet was resuspended in 40 μl of SDS sample buffer. A 20-μl aliquot of each sample was then resolved by SDS–PAGE using a 4–20% gradient gel, and EIIA$^{Glc}$ was visualized by western blot using anti-EIIA$^{Glc}$ serum[11].

**Measurement of the cellular protein level by western blot.** A *V. cholerae* strain in which the chromosomal *vc1710* was tagged with 3xFLAG at its C-terminus was grown in LB medium and aliquots were harvested when OD$_{600}$ reached 0.4, 1.0, and 1.5, respectively. After the pellets were resuspended in SDS-loading buffer, cells were lysed by boiling for 5 min. Cell lysates were electrophoresed on a SDS–PAGE gel with various amounts of purified VC1710::3xFLAG protein or EIIA$^{Glc}$ as control. For immunodetection, monoclonal mouse anti-FLAG antibody (Sigma-Aldrich) or anti-EIIA$^{Glc}$ mouse serum was used. Uncropped blot images were provided as a source data file.

**Quantification of *V. cholerae* colonization in the fly intestine.** To establish whether sugars could affect the gut colonization of *V. cholerae* strains, fasted adult flies (*Drosophila melanogaster* w1118) were orally administered a 5% sugar solution containing ~10$^6$ cells μl$^{-1}$ of the *V. cholerae* strains in the presence of either glucose or mannitol for 24 h, and fed the same, but sterile, medium for another 9 h. After each fly was immersed in 70% ethanol for 3 min, the surface-sterilized fly was homogenized in 1 ml of sterile PBS and spread on LB agar plates containing 10 μg ml$^{-1}$ of streptomycin to determine CFUs of *V. cholerae*. To image bacterial infection, the whole alimentary canal was dissected at the same time point, and visualized by confocal laser scanning microscopy.

For measurement of food consumption, the feeding assay was performed with a standard method using FD&C Blue #1 dye[64]. Flies were orally administered a 5% sugar (glucose or mannitol) solution containing *V. cholerae* cells and 0.5% (w/v) FD&C #1 dye for 2 h. Guts dissected from five flies were homogenized in 200 μl of PBS and the homogenate were centrifuged at 8000×g for 10 min. Food consumption was quantified by measuring absorbance of the supernatant at 625 nm.

**Reporting summary.** Further information on research design is available in the Nature Research Reporting Summary linked to this article.

## Data availability
The source data underlying Figs. 1a–d, 2b–d, 3a, c, 4a, b, 5a, b, d, and Supplementary Figs. 1a, c, 2, 3, 4a, b, 5a–c, 6, 7a, b and 8, are provided as a Source Data file.

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

## Acknowledgements

This work was supported by National Research Foundation Grants NRF-2018R1A5A1025077 and NRF-2019R1A2C2004143 funded by the Ministry of Science and ICT. W.-J.L. and K.-A.L. are supported by the national creative research initiative program (2015R1A3A2033475) and basic science research program (NRF-2019R1I1A1A01059606) of the National Research Foundation of South Korea. K.H. and H.-I.H. were supported by the BK21 plus program of the National Research Foundation of South Korea.

## Author contributions

K.H., Y.-H.P., H.-I.H., and Y.-J.S. designed the study. K.H., Y.-H.P., K.-A.L., and J.K. performed the experiments. K.H., Y.-H.P., K.-A.L., B.-G.K., W.-J.L., and Y.-J.S. analyzed data. K.H., H.-I.H., W.-J.L. and Y.-J.S. wrote the paper.

## Competing interests

The authors declare no competing interests.
