## [Peer Review File · Nature Communications]

Reviewers' comments:

Reviewer #1 (Remarks to the Author):

The PEP:glycose phosphotransferase system (PTS) transports and phosphorylates a wide range of carbohydrates. In addition, it has been shown to regulate numerous important cellular functions in response to the availability of efficiently metabolizable carbon sources. The PTS was also suggested to affect the virulence of certain pathogens. Regulation by the PTS is usually mediated via the phosphorylation state of its protein components.

The authors of manuscript NCOMMS-19-17087-T add a highly interesting example to the long list of PTS-regulated cellular functions: The glucose-specific EIIA (EIIAGlc) of *Vibrio cholerae* was found to interact with the c-di-GMP phosphodiesterase PdeS. EIIAGlc inhibits PdeS activity when it is unphosphorylated and stimulates PdeS when it is phosphorylated (P~EIIAGlc). In the presence of glucose, *V. cholerae* contains mainly EIIAGlc, which inhibits the hydrolysis of c-di-GMP by PdeS and consequently leads to elevated biofilm formation compared to *V. cholerae* cells grown in LB medium without glucose (contain mainly P~EIIAGlc). Utilization of different carbon sources was ultimately found to affect intestinal colonization of the host in a *Drosophila melanogaster* infection model. Colonization was more efficient by glucose-grown *V. cholerae* compared to mannitol-grown cells, in which only about 50% of EIIAGlc is present in dephospho form (elevated PdeS activity, lower biofilm formation).

In conclusion, the authors provide sound evidence for an entire *V. cholerae* regulatory cascade leading from carbon source utilization (environmental signal) via alterations of the phosphorylation state of PTS proteins, changes of the activity of the c-di-GMP phosphodiesterase PdeS and altered biofilm formation ultimately to differences in intestinal *V. cholerae* colonization of *D. melanogaster*. Although the PTS had previously been suggested to regulate biofilm formation, the underlying mechanism via EIIAGlc-mediated regulation of PdeS activity was not known before. Similar mechanisms of biofilm formation are expected to be operative in other organisms and the work described here is therefore of great general interest. The presented results are supported by careful statistical analyses.

A few minor points which should be addressed by the authors:

l. 126: Include the references for the interaction of EIIAGlc with IDE and FapA

l. 168: Maybe better write: VC1710 exhibited the predicted c-di-GMP hydrolytic activity

l. 181: What are the "c-di-GMP-bound effectors"?

l. 236: "What are "environmental sugars" compared to non-environmental sugars? Do the authors mean sugars encountered by *V. cholerae* in their environmental niches? See also legend of Fig. 4., l. 556. Also note, succinate is not a sugar. The abbreviation Suc could be easily assumed to stand for the sugar sucrose.

l. 237:presence of various sugars

l. 345 : What do the authors mean with "transmission to nature"?

l. 387: "To test the effect of EIIAGlc on the PDE activity, EIIAGlc and a trace amount of EI and HPr were added together with either glucose to dephosphorylate or PEP to phosphorylate EIIAGlc." Under the described conditions, glucose will not dephosphorylate P~EIIAGlc. Dephosphorylation of P~EIIAGlc requires in addition the membrane protein PtsG. Did the authors include PtsG-containing membrane fragments in the assay mixture? In fact, is glucose-mediated dephosphorylation necessary? Purified EIIAGlc is probably present at 99% in dephosphorylated form. As long as no PEP is added, phosphorylation will not occur! The same point applies to the legend of Fig. 2. In addition, what was the purpose of adding glucose in Supplem. Fig. 3? Glucose should not affect PdeS activity; it arrives as Glc-6-P in the cell.

l. 440 and 459: What does "□" stand for?

Fig. 5: How often was the confocal laser scanning microscopy experiment of the infected *D. melanogaster* intestine repeated and were the results always similar to those presented in Fig. 5? Suppl. Fig. 2b: The authors present 3 symbols for VC1710 alone, EIIAGlc, and P~EIIAGlc.

However, in the figure I see only black and grey squares, but no white squares for unphosphorylated EIIAGlc.

Reviewer #2 (Remarks to the Author):

In the manuscript entitled "The sugar-mediated regulation of c-di-GMP phosphodiesterase in *Vibrio cholerae*" Heo et al describe a novel EIIAGlc binding partner encoded by the locus VC1710, which has an EAL domain and represses *V. cholerae* biofilm formation by degrading c-di-GMP. They name this protein PdeS. The results supporting the conclusion that PdeS is a phosphodiesterase that is activated by phosphorylated EIIAGlc and inhibited by dephosphorylated EIIAGlc are convincing. EIIAGlc is known to regulate cellular physiology through its interactions with multiple partners. Here the authors have identified a novel partner of EIIAGlc. However, there are discrepancies between these results and published results. These may simply be the result of strain differences as *V. cholerae* is known to undergo genetic drift in laboratory culture, and N16961 has been in the laboratory for decades [1]. This should be discussed. Furthermore, the design of the biofilm and *Drosophila* experiments raises some questions. Specific comments follow:

- 1) Line 57: Transport of sugars by the PTS has been comprehensively studied by the Dalia lab [2]. The authors should consider referencing this manuscript.
- 2) Lines 68-70: This statement suggests that only one component of the PTS regulates biofilm formation and that this component has not been identified. This statement does not faithfully represent the literature. From published work, there is strong evidence that phosphorylated Enzyme I represses biofilm formation [3, 4]. Furthermore, at least two partners of EIIAGlc have previously been shown to repress biofilm formation. These are MshH, the *E. coli* CsrD homolog that also contains GGDEF and EAL motifs, and adenylate cyclase [4-8]. It would be more appropriate to state that regulation of biofilm formation by the PTS has not been fully elucidated.
- 3) Lines 85-87: The authors suggest that no direct link between the PTS and biofilm formation has been established. Certainly, a regulatory link between these two processes has been established in the case of EIIAGlc partners adenylate cyclase and MshH/CsrD. The transcription factor CRP, which binds cAMP, has been shown to regulate transcription of diguanylate cyclases and phosphodiesterases [8].
- 4) Lines 103-106 and Supplementary Figure 1.
 - A) these data set up the entire manuscript. Therefore, I suggest they be moved to the main Figures.
 - B) These data do not seem to be consistent with previously published data showing that EIIAGlc activates biofilm formation [9]. However, the authors do not directly compare biofilm formation by wild-type *V. cholerae* strain and the *crr* mutant, and this is curious. Upon digging deeper, the authors report biofilm formation as biofilm quantification divided by planktonic growth, so the question arises as to whether these measurements cannot be compared because planktonic growth is different for the wild-type and mutant. If so, a growth curve should be shown to make clear what the differences are.
 - C) The authors divide biofilm formation by planktonic growth, presumably to account for growth differences. This is not the appropriate way to normalize for growth differences. If a strain makes a bigger biofilm, there will be less planktonic cells in the medium. Therefore, the planktonic growth measurements may reflect either a difference in growth or a difference in biofilm formation. Total growth (biofilm+planktonic) is a better method for quantifying growth differences.
- 5) Lines 106-108: The authors note that cAMP has no effect on biofilm formation. This is contrary to what has been published in the literature multiple times with different *V. cholerae* strains. What concentration of cAMP was tested? I was unable to find this easily in the Figure Legends or the Experimental Procedures. If an adequate amount of cAMP was used, then the absence of an effect of cAMP on biofilm formation could be the result of strain variation or genetic drift in the laboratory. This discrepancy should be discussed.

6) Lines 108-110: This rationale for follow-up is poorly supported because biofilm formation by WT *V. cholerae* and the crr mutant is not directly compared in Supplementary Figure 1.

7) Lines 111-116: This description of the procedure is unclear. In line 115, please add "purified" His-EIIAGlc, if this is, in fact, what was done. At what point in the procedure were PEP and glucose added? Before the lysate was made, after the lysate was made, or after the complexes had been pulled down on the affinity beads? Also, would one expect EIIAGlc protein partners to already be complexed to native EIIAGlc, since wild-type *V. cholerae* was used to prepare the lysates? How much His-EIIAGlc was added to the mixture and was this in excess compared to the native amounts of EIIAGlc in wild-type cells?

8) Although there have been screens of all the c-di-GMP modulatory enzymes in the *V. cholerae* genome [10-12], VC1710 has not appeared in any of these as a biofilm-regulatory protein. This likely also represents a strain difference. In fact, the sum total of what was published about VC1710 prior to this manuscript was that it bound c-di-GMP [13]. This should be discussed, and all these works should be referenced.

9) Line 249: The use of the *Drosophila* model to test the function of PdeS is innovative, but there are some questions about the experimental model.

A) The methods state that the flies were administered "a medium" containing either 5% glucose or mannitol, but the nature of this medium is not elucidated anywhere in the text. This should be clarified.

B) In previous publications, a concentration of approximately 10^8 *V. cholerae*/ml in LB has been used to infect *Drosophila*. Here, the authors use "medium containing 10^{11} *V. cholerae*." Because no denominator is given, volume or otherwise, it is impossible to compare this infection protocol to the more commonly used one. A denominator should be given.

C) In more conventional models, the infection has been allowed to proceed over at least 24 hours. Here, the authors give the flies access to medium containing *V. cholerae* for just two hours, and then "wash-out" the bacteria for 2 hours with sterile medium. This is an extremely accelerated experiment. It is not clear to me why the authors chose such a short time. Is it possible that the difference diminishes over time? This should be discussed.

D) The question arises of how consistent infection is being obtained in such a short period. Intake varies, especially in the short term, depending on whether the flies "like" the medium/bacteria or not. Therefore, the authors should measure total intake over this two hour period under each condition.

E) The authors observe a difference in *V. cholerae* colonization in the anterior midgut. This is in contrast to most infection models in which pathogens colonize the posterior midgut. Therefore, the authors could be observing ingested bacteria that are passing through. A good way to prove this is not the case is to normalize to passage of a fluorescent or colored food additive. For instance, showing that the additive is excreted, while *V. cholerae* remains.

F) It is possible that inappropriate biofilm formation in the anterior midgut is responsible for the differences observed by the authors. Good proof of this would be to test Δvps and $\Delta vps \Delta pdeS$ mutants as well. c-di-GMP controls many things besides biofilm formation.

1. Stutzmann, S. and Blokesch, M. (2016) Circulation of a Quorum-Sensing-Impaired Variant of *Vibrio cholerae* Strain C6706 Masks Important Phenotypes. *mSphere* 1 (3).

2. Hayes, C.A. et al. (2017) Systematic genetic dissection of PTS in *Vibrio cholerae* uncovers a novel glucose transporter and a limited role for PTS during infection of a mammalian host. *Mol Microbiol* 104 (4), 568-579.

3. Vijayakumar, V. et al. (2018) Removal of a Membrane Anchor Reveals the Opposing Regulatory Functions of *Vibrio cholerae* Glucose-Specific Enzyme IIA in Biofilms and the Mammalian Intestine. *MBio* 9 (5).

4. Houot, L. and Watnick, P.I. (2008) A novel role for enzyme I of the *Vibrio cholerae* phosphoenolpyruvate phosphotransferase system in regulation of growth in a biofilm. *J Bacteriol* 190 (1), 311-20.

5. Leng, Y. et al. (2016) Regulation of CsrB/C sRNA decay by EIIA(Glc) of the phosphoenolpyruvate: carbohydrate phosphotransferase system. *Mol Microbiol* 99 (4), 627-39.

6. Pickering, B.S. et al. (2012) Glucose-specific enzyme IIA has unique binding partners in the *vibrio cholerae* biofilm. *MBio* 3 (6), e00228-12.

7. Liang, W. et al. (2007) The cyclic AMP receptor protein modulates colonial morphology in *Vibrio cholerae*. *Appl Environ Microbiol* 73 (22), 7482-7.
8. Fong, J.C. and Yildiz, F.H. (2008) Interplay between cyclic AMP-cyclic AMP receptor protein and cyclic di-GMP signaling in *Vibrio cholerae* biofilm formation. *J Bacteriol* 190 (20), 6646-59.
9. Houot, L. et al. (2010) The phosphoenolpyruvate phosphotransferase system regulates *Vibrio cholerae* biofilm formation through multiple independent pathways. *J Bacteriol* 192 (12), 3055-67.
10. Lim, B. et al. (2006) Cyclic-diGMP signal transduction systems in *Vibrio cholerae*: modulation of rugosity and biofilm formation. *Mol Microbiol* 60 (2), 331-48.
11. Liu, X. et al. (2010) Identification and characterization of a phosphodiesterase that inversely regulates motility and biofilm formation in *Vibrio cholerae*. *J Bacteriol* 192 (18), 4541-52.
12. Lim, B. et al. (2007) Regulation of *Vibrio* polysaccharide synthesis and virulence factor production by CdgC, a GGDEF-EAL domain protein, in *Vibrio cholerae*. *J Bacteriol* 189 (3), 717-29.
13. Roelofs, K.G. et al. (2015) Systematic Identification of Cyclic-di-GMP Binding Proteins in *Vibrio cholerae* Reveals a Novel Class of Cyclic-di-GMP-Binding ATPases Associated with Type II Secretion Systems. *PLoS Pathog* 11 (10), e1005232.

Reviewer #3 (Remarks to the Author):

The work presented by Heo et al. demonstrates how the phosphoenolpyruvate phosphotransferase system (PTS), specifically the glucose-specific enzyme EIIA-Glc, regulates the activity of cyclic-di-GMP (c-di-GMP) phosphodiesterase (PdeS) in *Vibrio cholerae*. EIIA-Glc/PdeS interaction was found to be dependent on the phosphorylation status of EIIA-Glc. The authors also show that the EIIA-Glc/PdeS signaling module controls in vitro biofilm formation and colonization in a *Drosophila* model of infection.

While previous studies demonstrated that PTS systems regulate biofilm formation in *V. cholerae*, interplay between PTS systems and c-di-GMP signaling systems has not been studied. Therefore the work presented in this study is significant.

Comments:

- 1) It was shown previously that "regulation of biofilm formation by EIIAGlc does not require phosphorylation of the conserved histidine at position 91" DOI:10.1128/JB.00213-10. It is important to generate an EIIAGlc dephosphomimetic mutant on the chromosome and analyze PdeS interaction and biofilm formation. This also serves as a genetic control for the studies performed.
- 2) EIIAGlc interaction partners were reported previously DOI:10.1128/mBio.00228-12. Please incorporate this work into your discussion.
- 3) Please provide an explanation for the differences in biofilm formation by N16961 in Fig 3a and 3b; indicate the amount of inducer used and include no inducer control.
- 4) Please provide concentrations of EI and HPr used in the experiments. The description provided ("trace amounts") is not sufficient for reproducibility studies.

Dear Reviewers,

We really appreciate your constructive and invaluable suggestions to improve our manuscript. We tried to go over the manuscript carefully and make it clearer and more accessible to readers. In this version, we added more experimental data and modified the manuscript as recommended. All changes were highlighted in red in the manuscript.

In summary:

1. We changed all the normalized data for biofilm formation experiments (A_{550}/A_{600}) into unprocessed data (A_{550}) as suggested by reviewer 2. These include Figures 1a, 3a, 3c, and 4a and Supplementary Figures 6a and 7. Also, we compared the growth of all strains used in this study to show that there were no significant growth differences between the strains under experimental conditions.
2. As reviewer 2 suggested, we performed the *V. cholerae* colonization assay for a longer infection period and with a *vpsA* mutant to assess the role of VPS on the PdeS-mediated regulation of intestinal colonization (Figure 5a).
3. We measured the amount of food consumed by flies according to the supplemented sugar and the *V. cholerae* strain to normalize the bacterial colonization, as reviewer 2 suggested.
4. We generated a dephosphomimetic mutant (H91A) of EIIA^{Glc} on the chromosome and analyzed its effect on biofilm formation and its interaction with PdeS, as reviewer 3 suggested.
5. We included data for the biofilm assays performed in the absence of the inducer arabinose in Supplementary Figure 7b, as reviewer 3 recommended.
6. We addressed all other issues raised by the three reviewers and made corrections for trivial mistakes.

Point-to-point response to the reviewers' comments

Reviewer #1 (Remarks to the Author):

The PEP:glycose phosphotransferase system (PTS) transports and phosphorylates a wide range of carbohydrates. In addition, it has been shown to regulate numerous important cellular functions in response to the availability of efficiently metabolizable carbon sources. The PTS was also suggested to affect the virulence of certain pathogens. Regulation by the PTS is usually mediated via the phosphorylation state of its protein components.

The authors of manuscript NCOMMS-19-17087-T add a highly interesting example to the long list of PTS-regulated cellular functions: The glucose-specific EIIA (EIIAGlc) of *Vibrio cholerae* was found to interact with the c-di-GMP phosphodiesterase PdeS. EIIAGlc inhibits PdeS activity when it is unphosphorylated and stimulates PdeS when it is phosphorylated (P~EIIAGlc). In the presence of glucose, *V. cholerae* contains mainly EIIAGlc, which inhibits the hydrolysis of c-di-GMP by PdeS and consequently leads to elevated biofilm formation compared to *V. cholerae* cells grown in LB medium without glucose (contain mainly P~EIIAGlc). Utilization of different carbon sources was ultimately found to affect intestinal colonization of the host in a *Drosophila melanogaster* infection model. Colonization was more efficient by glucose-grown *V. cholerae* compared to mannitol-grown cells, in which only about 50% of EIIAGlc is present in dephospho form (elevated PdeS activity, lower biofilm formation).

In conclusion, the authors provide sound evidence for an entire *V. cholerae* regulatory cascade leading from carbon source utilization (environmental signal) via alterations of the phosphorylation state of PTS proteins, changes of the activity of the c-di-GMP phosphodiesterase PdeS and altered biofilm formation ultimately to differences in intestinal *V. cholerae* colonization of *D. melanogaster*. Although the PTS had previously been suggested to regulate biofilm formation, the underlying mechanism via EIIAGlc-mediated regulation of PdeS activity was not known before. Similar mechanisms of biofilm formation are expected to be operative in other organisms and the work described here is therefore of great general interest. The presented results are supported by careful statistical analyses.

→ **We really appreciate the reviewer's careful reading and good suggestions. We made changes according to the reviewer's suggestion.**

A few minor points which should be addressed by the authors:

1. 126: Include the references for the interaction of EIIAGlc with IDE and FapA

→ **We added the reference as suggested (line 138).**

2. 168: Maybe better write: VC1710 exhibited the predicted c-di-GMP hydrolytic activity

→ **We revised the sentence as suggested (lines 181-182).**

3. 181: What are the "c-di-GMP-bound effectors"?

→ **We corrected this mistake to "c-di-GMP-binding proteins" (line 194).**

4. 236: "What are "environmental sugars" compared to non-environmental sugars? Do the authors mean sugars encountered by *V. cholerae* in their environmental niches? See also legend of Fig. 4., l. 556. Also note, succinate is not a sugar. The abbreviation Suc could be easily assumed to stand for the sugar sucrose.

→ **We changed "environmental sugars" to "carbon sources" (lines 26, 99, and 256 and the legend to Figure 4) or to "carbohydrates encountered by *V. cholerae* in their environmental niches" (lines 260-261) and changed "Suc" to "Succ" in Figure 4 and Supplementary Figure 8.**

5. 237:presence of various sugars

→ **We changed "various sugar" to "various carbon sources" (line 262).**

6. 345 : What do the authors mean with "transmission to nature"?

→ **We changed the sentence to "... and transmission to new hosts" (line 395).**

7. 387: "To test the effect of EIIAGlc on the PDE activity, EIIAGlc and a trace amount of EI and HPr were added together with either glucose to dephosphorylate or PEP to phosphorylate EIIAGlc." Under the described conditions, glucose will not dephosphorylate P~EIIAGlc. Dephosphorylation of P~EIIAGlc requires in addition the membrane protein PtsG. Did the

authors include PtsG-containing membrane fragments in the assay mixture? In fact, is glucose-mediated dephosphorylation necessary? Purified EIIAGlc is probably present at 99% in dephosphorylated form. As long as no PEP is added, phosphorylation will not occur! The same point applies to the legend of Fig. 2. In addition, what was the purpose of adding glucose in Supplem. Fig. 3? Glucose should not affect PdeS activity; it arrives as Glc-6-P in the cell.

→ **We agree with the reviewer and we corrected our mistakes. Because we overexpressed and purified PTS components including His-EIIA^{Glc} in a *ptsH/crr*-deleted ER2566 strain, HPr and EIIA^{Glc} were purified in their dephosphorylated forms. While it was necessary to add glucose in Figure 1 to completely dephosphorylate His-EIIA^{Glc} and the other PTS components in the presence of the wild-type *V. cholerae* extract, we did not have to add glucose to the reaction mixtures of purified proteins to dephosphorylate them. Therefore, we corrected these mistakes in the Methods section (lines 436-438), Figure 2 and Supplementary Figure 3. We described the methods for protein purification in more detail in the revised version (lines 406-408).**

8. 440 and 459: What does “□” stand for?

→ **We made corrections (lines 489 and 507).**

Fig. 5: How often was the confocal laser scanning microscopy experiment of the infected *D. melanogaster* intestine repeated and were the results always similar to those presented in Fig. 5?

→ **We repeated the confocal laser scanning microscopy experiment three times on different days, and we always obtained similar results to Figure 5c. This was mentioned in the revised version.**

Suppl. Fig. 2b: The authors present 3 symbols for VC1710 alone, EIIAGlc, and P~EIIAGlc. However, in the figure I see only black and grey squares, but no white squares for unphosphorylated EIIAGlc.

→ **We presented symbols for experimental data in each panel.**

Reviewer #2 (Remarks to the Author):

In the manuscript entitled “The sugar-mediated regulation of c-di-GMP phosphodiesterase in *Vibrio cholerae*” Heo et al describe a novel EllAGlc binding partner encoded by the locus VC1710, which has an EAL domain and represses *V. cholerae* biofilm formation by degrading c-di-GMP. They name this protein PdeS. The results supporting the conclusion that PdeS is a phosphodiesterase that is activated by phosphorylated EllAGlc and inhibited by dephosphorylated EllAGlc are convincing. EllAGlc is known to regulate cellular physiology through its interactions with multiple partners. Here the authors have identified a novel partner of EllAGlc. However, there are discrepancies between these results and published results. These may simply be the result of strain differences as *V. cholerae* is known to undergo genetic drift in laboratory culture, and N16961 has been in the laboratory for decades [1]. This should be discussed. Furthermore, the design of the biofilm and *Drosophila* experiments raises some questions. Specific comments follow:

→ **Thank you for the invaluable comments and suggestions.**

1) Line 57: Transport of sugars by the PTS has been comprehensively studied by the Dalia lab [2]. The authors should consider referencing this manuscript.

→ **The reference was cited in the manuscript (line 57).**

2) Lines 68-70: This statement suggests that only one component of the PTS regulates biofilm formation and that this component has not been identified. This statement does not faithfully represent the literature. From published work, there is strong evidence that phosphorylated Enzyme I represses biofilm formation [3, 4]. Furthermore, at least two partners of EllAGlc have previously been shown to repress biofilm formation. These are MshH, the *E. coli* CsrD homolog that also contains GGDEF and EAL motifs, and adenylate cyclase [4-8]. It would be more appropriate to state that regulation of biofilm formation by the PTS has not been fully elucidated.

→ **We revised this part as suggested, referring to the previously reported binding partners of EllA^{Glc} (lines 69-70, 86-97).**

3) Lines 85-87: The authors suggest that no direct link between the PTS and biofilm formation has been established. Certainly, a regulatory link between these two processes has been established in the case of EllAGlc partners adenylate cyclase and MshH/CsrD. The transcription factor CRP, which binds cAMP, has been shown to regulate transcription of diguanylate cyclases and phosphodiesterases [8].

→ We rephrased this part as follows: “The regulatory functions of EIIA^{Glc} on biofilm formation were suggested in *V. cholerae* in previous studies. It was reported that cAMP, the reaction product of adenylate cyclase which is regulated by EIIA^{Glc}, and its receptor protein (CRP) directly and indirectly represses the expression of the diguanylate cyclase CdgA, which positively regulates biofilm formation in the *V. cholerae* C6706 strain^{1,2}. However, in the *V. cholerae* MO10 strain, while EIIA^{Glc} activated cell growth in biofilm formation in the minimal medium supplemented with glucose, exogenous addition of cAMP to 2 mM had no effect on total growth and biofilm accumulation by a *crr* mutant³. EIIA^{Glc} was also shown to interact with MshH and activate biofilm formation⁴. While it was proposed that MshH may function upstream of EIIA^{Glc} in the regulation of biofilm formation, how EIIA^{Glc} activates biofilm formation is yet unclear. Therefore, it is puzzling how EIIA^{Glc} regulates biofilm formation in *V. cholerae*” (lines 86-97).

4) Lines 103-106 and Supplementary Figure 1.

A) these data set up the entire manuscript. Therefore, I suggest they be moved to the main Figures.

→ We moved Supplementary Figure 1 to the main Figure 1a, as suggested.

B) These data do not seem to be consistent with previously published data showing that EllAGlc activates biofilm formation [9]. However, the authors do not directly compare biofilm formation by wild-type *V. cholerae* strain and the *crr* mutant, and this is curious. Upon digging deeper, the authors report biofilm formation as biofilm quantification divided

by planktonic growth, so the question arises as to whether these measurements cannot be compared because planktonic growth is different for the wild-type and mutant. If so, a growth curve should be shown to make clear what the differences are.

→ **We compared biofilm formation by wild-type *V. cholerae* and its *crr* mutant in LB medium, as shown in Figure 1a, and we added the following sentence: “Interestingly, while the *crr* mutant had a similar growth (Supplementary Fig. 1), this mutant exhibited a higher level of biofilm formation compared to the wild-type strain in LB medium (Fig. 1a), which is contrary to a previous study⁵.” (lines 116-119)**

C) The authors divide biofilm formation by planktonic growth, presumably to account for growth differences. This is not the appropriate way to normalize for growth differences. If a strain makes a bigger biofilm, there will be less planktonic cells in the medium. Therefore, the planktonic growth measurements may reflect either a difference in growth or a difference in biofilm formation. Total growth (biofilm+planktonic) is a better method for quantifying growth differences.

→ **To avoid any possible complication due to different growth rates among strains, we measured the total growth and the amount of biofilm formation separately (See figures 1a, 3a, 3c and 4 and Supplementary Figures 1, 6a and 7).**

5) Lines 106-108: The authors note that cAMP has no effect on biofilm formation. This is contrary to what has been published in the literature multiple times with different *V. cholerae* strains. What concentration of cAMP was tested? I was unable to find this easily in the Figure Legends or the Experimental Procedures. If an adequate amount of cAMP was used, then the absence of an effect of cAMP on biofilm formation could be the result of strain variation or genetic drift in the laboratory. This discrepancy should be discussed.

→ **We added 5 mM cAMP into the medium, which was previously shown to be an effective concentration⁶. In previous studies, different effects of cAMP-CRP on biofilm formation have been reported among various *V. cholerae* strains. While biofilm formation was negatively regulated by the cAMP-CRP complex in the C6706**

and C6728 strains^{2,7}, the supplementation of the growth medium with various concentrations of cAMP had no effect on the total growth and biofilm accumulation by a *crr* mutant of the MO10 strain³. Herein we report that the sugar-dependent regulation of biofilm formation is hardly affected by cAMP-CRP in the N16961 strain. This may simply be the result of strain differences, as *V. cholerae* is known to undergo genetic drift in laboratory culture⁸. We discussed about these differences in the revised manuscript, as the reviewer suggested (lines 382-390).

6) Lines 108-110: This rationale for follow-up is poorly supported because biofilm formation by WT *V. cholerae* and the *crr* mutant is not directly compared in Figure 1a.

→ **We compared the biofilm formation of the two strains in the revised manuscript (lines 116-119).**

7) Lines 111-116: This description of the procedure is unclear. In line 115, please add “purified” His-EllAGlc, if this is, in fact, what was done. At what point in the procedure were PEP and glucose added? Before the lysate was made, after the lysate was made, or after the complexes had been pulled down on the affinity beads? Also, would one expect EllAGlc protein partners to already be complexed to native EllAGlc, since wild-type *V. cholerae* was used to prepare the lysates? How much His-EllAGlc was added to the mixture and was this in excess compared to the native amounts of EllAGlc in wild-type cells?

→ We added glucose or PEP to the mixture of the cell lysate, Talon metal affinity resin and 100 µg of purified His-EIIA^{Glc}, which is in excess of the amount of native EIIA^{Glc}

Figure 1a in Park *et al.*, 2019

Figure 1b in Park *et al.*, 2013

present in the wild-type cell lysate, and this mixture was then subjected to the pull-down experiment. This was clarified in the revised manuscript (lines 127, 418-419, 422). This way, we could successfully manipulate the phosphorylation state of the PTS components in the presence of either the *V. vulnificus* or *E. coli* cell extract in our previous studies^{9,10} (Figure 1a in Park *et al.*, 2019; Figure 1b in Park *et al.*, 2013).

8) Although there have been screens of all the c-di-GMP modulatory enzymes in the *V. cholerae* genome [10-12], VC1710 has not appeared in any of these as a biofilm-regulatory protein. This likely also represents a strain difference. In fact, the sum total of what was published about VC1710 prior to this manuscript was that it bound c-di-GMP [13]. This should be discussed, and all these works should be referenced.

→ We added the information as suggested (lines 176-177).

9) Line 249: The use of the *Drosophila* model to test the function of PdeS is innovative, but there are some questions about the experimental model.

A) The methods state that the flies were administered “a medium” containing either 5% glucose or mannitol, but the nature of this medium is not elucidated anywhere in the text. This should be clarified.

→ We made clarifications as suggested (lines 295-296, 314, 316, 524, and 533 and the legend to Figure 5).

B) In previous publications, a concentration of approximately 10^8 V. cholerae/ml in LB has been used to infect *Drosophila*. Here, the authors use “medium containing 10^{11} V. cholerae.” Because no denominator is given, volume or otherwise, it is impossible to compare this infection protocol to the more commonly used one. A denominator should be given.

→ We made corrections as suggested (lines 296, 314-315 and 524).

C) In more conventional models, the infection has been allowed to proceed over at least 24 hours. Here, the authors give the flies access to medium containing V. cholerae for just two hours, and then “wash-out” the bacteria for 2 hours with sterile medium. This is an extremely accelerated experiment. It is not clear to me why the authors chose such a short time. Is it possible that the difference diminishes over time? This should be discussed.

→ It is generally accepted that colonization of the *Drosophila* gut even by commensal bacteria is not so stable to be maintained for a long time. This is supported by the fact that consecutive transfers of flies to germ-free food vials dramatically reduce the number of commensal bacteria following each transfer, ultimately resulting in an almost complete loss of gut commensal bacteria^{11,12}. Therefore, we assumed that the effects of PdeS and c-di-GMP on the attachment and colonization may be more evident during the early stage of infection due to a weak association of bacteria with epithelial cells of the *D. melanogaster* intestine. During this revision, however, we repeated the colonization assays for 24 hours and we confirmed that the inhibitory effect of PdeS on colonization could be observed also during this longer infection period. These results are added and discussed in the revised manuscript (Figure 5a).

D) The question arises of how consistent infection is being obtained in such a short period. Intake varies, especially in the short term, depending on whether the flies “like” the

medium/bacteria or not. Therefore, the authors should measure total intake over this two hour period under each condition.

→ **We quantified food consumption using FD&C Blue #1 dye during bacterial infection as suggested, and we added this result in the revised manuscript (Figure 5d).**

E) The authors observe a difference in *V. cholerae* colonization in the anterior midgut. This is in contrast to most infection models in which pathogens colonize the posterior midgut. Therefore, the authors could be observing ingested bacteria that are passing through. A good way to prove this is not the case is to normalize to passage of a fluorescent or colored food additive. For instance, showing that the additive is excreted, while *V. cholerae* remains.

→ **The *Drosophila* gastrointestinal tract consists of anterior midgut (AM), middle midgut (MM) with copper cell region (CCR), and posterior midgut (PM). CCR forms an acidic zone, and acts as a stomach-like function¹³. Due to the acidic zone between the AM and the PM, we and others found that ingested bacteria including commensal bacteria are predominantly localized and observed in the AM region rather than the PM region^{13,14}. In a meanwhile, we measured food consumption during *V. cholerae* infection experiments as suggested, and we observed that the amounts of ingested bacteria (as estimated by the ingested amounts of bacteria-containing sugar solutions) were similar between wild-type and $\Delta pdeS$ strains. Taken together, we believe that the GFP signals that we observed 2 hours after ingestion of bacteria reflect the colonization of bacteria on epithelia of the AM region. This result was added in the revised manuscript (Figure 5d).**

F) It is possible that inappropriate biofilm formation in the anterior midgut is responsible for the differences observed by the authors. Good proof of this would be to test Δvps and $\Delta vps \Delta pdeS$ mutants as well. c-di-GMP controls many things besides biofilm formation.

→ **We constructed $\Delta vpsA$ and $\Delta vpsA pdeS$ mutants, and repeated the colonization assay of Figure 5a to determine whether VPS synthesis and biofilm formation were responsible for the difference in *V. cholerae* colonization in the anterior midgut, as deletion of *vpsA* was shown to result in significantly decreased VPS synthesis and hence biofilm formation¹⁵. We added a new paragraph describing the result as**

follows: “To determine whether the effect of PdeS on intestinal colonization is mediated by the regulation of biofilm formation, we repeated the colonization experiment with $\Delta vpsA$ and $\Delta vpsA pdeS$ mutants (Fig. 5a). As reported previously^{16,17}, mutation of *vpsA* resulted in a significantly reduced (~5%) intestinal colonization in flies compared to the wild-type strain. Interestingly, the inhibitory effect of PdeS on intestinal colonization was not seen in this *vpsA* mutant strain. Thus, it could be assumed that VPS-dependent biofilm formation is important for the regulation of intestinal colonization by PdeS.” (lines 306-312).

Reviewer #3 (Remarks to the Author):

The work presented by Heo et al. demonstrates how the phosphoenolpyruvate phosphotransferase system (PTS), specifically the glucose-specific enzyme EIIA-Glc, regulates the activity of cyclic-di-GMP (c-di-GMP) phosphodiesterase (PdeS) in *Vibrio cholerae*. EIIA-Glc/PdeS interaction was found to be dependent on the phosphorylation status of EIIA-Glc. The authors also show that the EIIA-Glc/PdeS signaling module controls in vitro biofilm formation and colonization in a *Drosophila* model of infection.

While previous studies demonstrated that PTS systems regulate biofilm formation in *V. cholerae*, interplay between PTS systems and c-di-GMP signaling systems has not been studied. Therefore the work presented in this study is significant.

Comments:

1) It was shown previously that “regulation of biofilm formation by EIIAGlc does not require phosphorylation of the conserved histidine at position 91” DOI:10.1128/JB.00213-10. It is important to generate an EIIAGlc dephosphomimetic mutant on the chromosome and analyze PdeS interaction and biofilm formation. This also serves as a genetic control for the studies performed.

→ **We constructed a dephosphomimetic mutant (H91A) of *crr* encoding EIIA^{Glc} on the chromosome and performed biofilm formation assays. And we added the results (Supplementary Figure 7a) and modified the paragraph as follows: “Then, to confirm whether the regulation of the PdeS activity by EIIA^{Glc} also operates in *V. cholerae* cells, we constructed a dephosphomimetic mutant (H91A) of *crr* encoding**

EIIA^{Glc} on the chromosome and performed biofilm formation assays (Supplementary Fig. 7a). While we observed that the dephosphomimetic *crr* mutant showed increased biofilm formation compared to the wild-type strain, this stimulatory effect was not observed in a *pdeS* deletion mutant. EIIA^{Glc} is shared for several membrane-bound enzyme IIBCs including those specific for glucose, *N*-acetylglucosamine, sucrose, and trehalose in *V. cholerae*⁵ and phosphorylatable EIIA^{Glc} is indispensable for a variety of physiological processes including the regulation of global transcription factors such as CRP and Mlc. Therefore, we assume that the phenotype of this chromosomal *crr* mutant might be due to indirect pleiotropic effects of the mutation. For this reason, we tried to see the effect of EIIA^{Glc} dephosphorylation on biofilm formation by increasing the level of the dephosphomimetic mutant of EIIA^{Glc} while minimizing the perturbation of the overall PTS activity. Therefore, we compared biofilm formation and the intracellular level of c-di-GMP of the wild-type strain carrying an expression vector for wild-type EIIA^{Glc} with the wild-type strain carrying an expression vector for EIIA^{Glc}(H91A), and observed increased biofilm formation and the c-di-GMP level in the latter strain (Fig. 3c), which is in accordance with the result obtained by the chromosomal mutation of the *crr* gene. Together our data show that dephosphorylated EIIA^{Glc} inactivates the PDE activity of PdeS and thereby increases the c-di-GMP level *in vivo*. Thus, it could be assumed that EIIA^{Glc} can modulate biofilm formation by controlling the c-di-GMP level through direct interaction with PdeS.” (lines 233-254)

2) EIIAGlc interaction partners were reported previously DOI:10.1128/mBio.00228-12. Please incorporate this work into your discussion.

→ We revised the manuscript as suggested (line 86-97).

3) Please provide an explanation for the differences in biofilm formation by N16961 in Fig 3a and 3b; indicate the amount of inducer used and include no inducer control.

→ We added sentences explaining for the differences in biofilm in Figure 3a and 3b (line 498-499). We used 0.1% arabinose to induce EIIA^{Glc} expression and we added the information in the revised manuscript (Figure 3c). We also included data for the

biofilm assays performed in the absence of the inducer arabinose in Supplementary Figure 7b.

4) Please provide concentrations of EI and HPr used in the experiments. The description provided (“trace amounts”) is not sufficient for reproducibility studies.

→ **We specified exact concentrations of EI and HPr in the revised manuscript (lines 437).**

References

- 1 Beyhan, S., Bilecen, K., Salama, S. R., Casper-Lindley, C. & Yildiz, F. H. Regulation of rugosity and biofilm formation in *Vibrio cholerae*: comparison of VpsT and VpsR regulons and epistasis analysis of *vpsT*, *vpsR*, and *hapR*. *J Bacteriol* **189**, 388-402, doi:10.1128/JB.00981-06 (2007).
- 2 Fong, J. C. & Yildiz, F. H. Interplay between cyclic AMP-cyclic AMP receptor protein and cyclic di-GMP signaling in *Vibrio cholerae* biofilm formation. *J Bacteriol* **190**, 6646-6659, doi:10.1128/JB.00466-08 (2008).
- 3 Houot, L. & Watnick, P. I. A novel role for enzyme I of the *Vibrio cholerae* phosphoenolpyruvate phosphotransferase system in regulation of growth in a biofilm. *J Bacteriol* **190**, 311-320, doi:10.1128/JB.01410-07 (2008).
- 4 Pickering, B. S., Smith, D. R. & Watnick, P. I. Glucose-specific enzyme IIA has unique binding partners in the *vibrio cholerae* biofilm. *MBio* **3**, e00228-00212, doi:10.1128/mBio.00228-12 (2012).
- 5 Houot, L., Chang, S., Absalon, C. & Watnick, P. I. *Vibrio cholerae* phosphoenolpyruvate phosphotransferase system control of carbohydrate transport, biofilm formation, and colonization of the germfree mouse intestine. *Infect Immun* **78**, 1482-1494, doi:10.1128/IAI.01356-09 (2010).
- 6 Wu, R. *et al.* Direct regulation of the natural competence regulator gene *tfoX* by cyclic AMP (cAMP) and cAMP receptor protein (CRP) in *Vibrios*. *Sci Rep* **5**, 14921, doi:10.1038/srep14921 (2015).
- 7 Liang, W., Silva, A. J. & Benitez, J. A. The cyclic AMP receptor protein modulates colonial morphology in *Vibrio cholerae*. *Appl Environ Microbiol* **73**, 7482-7487, doi:10.1128/AEM.01564-07 (2007).
- 8 Stutzmann, S. & Blokesch, M. Circulation of a Quorum-Sensing-Impaired Variant of *Vibrio cholerae* Strain C6706 Masks Important Phenotypes. *mSphere* **1**, doi:10.1128/mSphere.00098-16 (2016).
- 9 Park, S. *et al.* Polar landmark protein HubP recruits flagella assembly protein FapA under glucose limitation in *Vibrio vulnificus*. *Mol Microbiol* **112**, 266-279, doi:10.1111/mmi.14268 (2019).
- 10 Park, Y. H., Lee, C. R., Choe, M. & Seok, Y. J. HPr antagonizes the anti-sigma70 activity of Rsd in *Escherichia coli*. *Proc Natl Acad Sci U S A* **110**, 21142-21147, doi:10.1073/pnas.1316629111 (2013).
- 11 Blum, J. E., Fischer, C. N., Miles, J. & Handelsman, J. Frequent replenishment sustains the beneficial microbiome of *Drosophila melanogaster*. *MBio* **4**, e00860-00813, doi:10.1128/mBio.00860-13 (2013).
- 12 Broderick, N. A., Buchon, N. & Lemaitre, B. Microbiota-induced changes in *drosophila melanogaster* host gene expression and gut morphology. *MBio* **5**, e01117-01114, doi:10.1128/mBio.01117-14 (2014).
- 13 Li, H., Qi, Y. & Jasper, H. Preventing Age-Related Decline of Gut Compartmentalization Limits Microbiota Dysbiosis and Extends Lifespan. *Cell Host Microbe* **19**, 240-253, doi:10.1016/j.chom.2016.01.008 (2016).
- 14 Lee, K. A. *et al.* Bacterial-derived uracil as a modulator of mucosal immunity and gut-microbe homeostasis in *Drosophila*. *Cell* **153**, 797-811, doi:10.1016/j.cell.2013.04.009 (2013).
- 15 Fong, J. C., Syed, K. A., Klose, K. E. & Yildiz, F. H. Role of *Vibrio* polysaccharide (*vps*) genes in VPS production, biofilm formation and *Vibrio cholerae* pathogenesis. *Microbiology* **156**, 2757-2769, doi:10.1099/mic.0.040196-0 (2010).
- 16 Kamareddine, L. *et al.* Activation of *Vibrio cholerae* quorum sensing promotes survival of an arthropod host. *Nat Microbiol* **3**, 243-252, doi:10.1038/s41564-017-0065-7 (2018).
- 17 Purdy, A. E. & Watnick, P. I. Spatially selective colonization of the arthropod intestine through activation of *Vibrio cholerae* biofilm formation. *Proc Natl Acad Sci U S A* **108**, 19737-19742, doi:10.1073/pnas.1111530108 (2011).

REVIEWERS' COMMENTS:

Reviewer #2 (Remarks to the Author):

The authors are to be commended for their revisions, which have resulted in a greatly improved and convincing manuscript. There are just a few corrections that should be made. First, a comment on the big picture. EIIAGlc has many interacting partners, and several of them regulate biofilm formation. The role of EIIAGlc in biofilm formation has been studied in many different strains of *V. cholerae*. In each strain and under each condition tested, the observed regulatory role of EIIAGlc in biofilm formation depends on which partners are present and active. Therefore, the differing results observed here and by previous investigators do not conflict, but rather reflect true differences between strains. I think it is important to phrase this in such a way that the reader is not left with the impression that these findings are contradictory. I find them to be complementary.

1) Line 92: I find it curious that the authors comment on the effect of cAMP on biofilm formation by a *crr* mutant in the referenced publication. The *crr* mutant makes very little biofilm, so it would be difficult to measure a reduction with cAMP supplementation. On the other hand, similar to what has been reported for other wild-type *V. cholerae* strains, cAMP represses biofilm formation by the wild-type parental strain in this publication.

2) Line 93: This is incorrect. First of all, the referenced publication reports that MshH represses biofilm formation. Furthermore, a recent publication by the Romeo lab shows that EIIAGlc activates degradation of sRNAs by MshH. A subsequent publication by another lab probes this further. This puts EIIAGlc at the same level as MshH. MshH does NOT act upstream of EIIAGlc. It is most likely that CsrA, which is downstream of MshH, is the direct inhibitor of biofilm formation as it is in *V. vulnificus* (see Jones, AEM, 2008).

3) Lines 95-96: This is awkward and unclear. Rather than saying that it is puzzling, I suggest a statement to the effect that regulation of biofilm formation by EIIAGlc is complex and the sum of the action of multiple EIIAGlc partners. I suspect that in N16961 under the given conditions, only VC1710 is active, and, therefore, the EIIAGlc biofilm phenotype is solely due to this protein.

4) Rebuttal: The authors comment that commensal bacteria are only transiently associated with the fly intestine and furthermore that they are found in the anterior midgut due to the acidity of the middle midgut, which creates a bottleneck. While this is true, there is ample published evidence to show that, during longer infections, wild-type *V. cholerae* accesses the posterior midgut, forms a biofilm, and is able to stably colonize and grow in this compartment. This might be observed if one performed microscopy at 24 hours. However, I do not believe this experiment is necessary for this manuscript.

Reviewer #3 (Remarks to the Author):

This study provides new insights into mechanisms of environmental regulation of biofilm formation and c-di-GMP signaling in *Vibrio cholerae*. The revised manuscript is much improved, the results are clearly presented and conclusions are supported by the data.

Minor comment:

Line 90 should read "in the *V. cholerae* strain A1552" not "C6706".

Dear reviewers,

We really appreciate your careful reading and constructive suggestions to improve our manuscript. In this version, we modified the manuscript as recommended.

Point-to-point response to the reviewers' comments

Reviewer #2 (Remarks to the Author):

The authors are to be commended for their revisions, which have resulted in a greatly improved and convincing manuscript. There are just a few corrections that should be made. First, a comment on the big picture. EllAGlc has many interacting partners, and several of them regulate biofilm formation. The role of EllAGlc in biofilm formation has been studied in many different strains of *V. cholerae*. In each strain and under each condition tested, the observed regulatory role of EllAGlc in biofilm formation depends on which partners are present and active. Therefore, the differing results observed here and by previous investigators do not conflict, but rather reflect true differences between strains. I think it is important to phrase this in such a way that the reader is not left with the impression that these findings are contradictory. I find them to be complementary.

1) Line 92: I find it curious that the authors comment on the effect of cAMP on biofilm formation by a *crr* mutant in the referenced publication. The *crr* mutant makes very little biofilm, so it would be difficult to measure a reduction with cAMP supplementation. On the other hand, similar to what has been reported for other wild-type *V. cholerae* strains, cAMP represses biofilm formation by the wild-type parental strain in this publication.

→ **We agree that the effect of cAMP on biofilm formation was undetectable probably due to a very little amount of biofilm formation by the *crr* deletion mutant, although the authors in the cited publication (Houot and Watnick, 2008) described that supplementation of the growth medium with various concentrations of cAMP from 0 to 2 mM had no effect on biofilm accumulation by the Δcrr mutant (the second paragraph of the last subtitle on the right column of p. 318). Thus, we rephrased this**

part to: “While the exogenous addition of cAMP represses the biofilm formation also in the *V. cholerae* MO10 strain, EIIA^{Glc} was shown to activate biofilm formation in the presence of exogenous cAMP in this strain.” (lines 87-90)

2) Line 93: This is incorrect. First of all, the referenced publication reports that MshH represses biofilm formation. Furthermore, a recent publication by the Romeo lab shows that EllAGlc activates degradation of sRNAs by MshH. A subsequent publication by another lab probes this further. This puts EllAGlc at the same level as MshH. MshH does NOT act upstream of EllAGlc. It is most likely that CsrA, which is downstream of MshH, is the direct inhibitor of biofilm formation as it is in *V. vulnificus* (see Jones, AEM, 2008).

→ **We made corrections and added related information (lines 81-84)**

3) Lines 95-96: This is awkward and unclear. Rather than saying that it is puzzling, I suggest a statement to the effect that regulation of biofilm formation by EllAGlc is complex and the sum of the action of multiple EllAGlc partners. I suspect that in N16961 under the given conditions, only VC1710 is active, and, therefore, the EllAGlc biofilm phenotype is solely due to this protein.

→ **We rephrased this part as suggested, showing that biofilm formation might be regulated in combination with actions by multiple partners of EIIA^{Glc} (lines 80-91).**

4) Rebuttal: The authors comment that commensal bacteria are only transiently associated with the fly intestine and furthermore that they are found in the anterior midgut due to the acidity of the middle midgut, which creates a bottleneck. While this is true, there is ample published evidence to show that, during longer infections, wild-type *V. cholerae* accesses the posterior midgut, forms a biofilm, and is able to stably colonize and grow in this compartment. This might be observed if one performed microscopy at 24 hours. However, I do not believe this experiment is necessary for this manuscript.

→ **We totally agree with the reviewer’s points. In fact, colonization patterns vary greatly depending on different experimental conditions; for example, (1) infection doses (i.e. bacterial CFU used for the infection), (2) duration of infection, (3) bacterial species used for the experiment, and (4) host physiology (age and sex of the host animals). Therefore, to avoid any confusion, we have added ‘in our infection condition’ in the text when we describe the bacterial colonization in the anterior midgut (line 320).**

Reviewer #3 (Remarks to the Author):

This study provides new insights into mechanisms of environmental regulation of biofilm formation and c-di-GMP signaling in *Vibrio cholerae*. The revised manuscript is much improved, the results are clearly presented and conclusions are supported by the data.

Minor comment:

Line 90 should read “in the *V. cholerae* strain A1552” not “C6706”.

→ **We made corrections as suggested (lines 90, 375).**